# Warm-Up Strategies at Halftime: A Pilot Randomized Controlled Trial in a Professional Women’s Soccer Team

**DOI:** 10.3390/jfmk10030270

**Published:** 2025-07-16

**Authors:** Marco Abreu, Fábio Y. Nakamura, Thiago Carvalho, Davi Silva, Fabrício Vasconcellos, José Afonso

**Affiliations:** 1Centre of Research, Education, Innovation, and Intervention in Sport (CIFI_2_D), Faculty of Sport, University of Porto (FADEUP), 4200-450 Porto, Portugal; profmarcoabreu@hotmail.com (M.A.); up201005205@up.pt (T.C.); 2Research Center in Sports Sciences, Health Sciences and Human Development (CIDESD), University of Maia, 4475-690 Maia, Portugal; fabioy_nakamura@yahoo.com.br; 3Post-Graduate Program in Exercise and Sport Sciences, Laboratory of Soccer Studies (LABESFUT), Rio de Janeiro State University, Rio de Janeiro 22763010, Brazil; davizirt@hotmail.com (D.S.); fabricio.vasconcellos@uerj.br (F.V.)

**Keywords:** re-warm-up, halftime, rest period, jump, sprint, performance

## Abstract

**Objectives:** We compared the effects of two active re-warm-up protocols applied during halftime’s last three minutes, after a warm-up, testing, and a simulated first-half match. **Methods:** Twenty-two professional players from a first Portuguese division club were randomized into two re-warm-up protocols during a simulated match interval: (i) a strength, plyometrics, and balance protocol (SPBP); and (ii) a soccer-specific protocol (SSP). Players were assessed for a 20-m linear sprint and countermovement jump (CMJ) after the warm-up and the re-warm-up. Descriptive statistics and mixed ANOVA were performed, with effect size assessed using partial eta-squared. The Acute Readiness Monitoring Scale (ARMS) questionnaire was administered after the simulated match and re-warm-up and was analyzed using a multifactorial ANOVA. **Results:** No significant interaction effects were observed (*p* > 0.05). Comparing pre-match to post-re-warm-up, there was a slight decrease in sprint (significant) and jump performance (non-significant). Additionally, there were no between-protocol differences in perceived readiness (ARMS). **Conclusions:** After the three-minute re-warm-up protocols, similar results were observed in the 20-m sprint performance, CMJ, and perceived readiness when comparing SPBP and SSP. These re-warm-up protocols (SPBP and SSP) are practical to implement within a 3-min time window, and, given their apparent lack of differences, players’ preferences could be considered. However, the SSP is currently subject to restrictions that limit teams’ access to the field during this period. Future research should compare active re-warm-up protocols with passive controls to more clearly assess their effectiveness.

## 1. Introduction

Warm-up practices are fundamental in team sports, offering physiological and physical benefits [1], preparing athletes for optimal performance [2]. They enhance athletic performance by increasing muscle temperature, accelerating metabolism, optimizing force/power output, and neuromuscular activation [3,4]. Warm-ups accelerate and optimize VO_2_ kinetics to more efficiently meet exercise demands [5,6], activate the central nervous system [7], and improve performance in sprinting and changes of direction (COD) tasks [8]. Warm-ups are applied in various contexts: pre-training, pre-match, halftime, and during the match [9]. While the traditional focus has been on pre-training or pre-match warm-ups, increasing attention is now being paid to re-warm-up strategies during halftime.

Prolonged passive breaks during halftime can negatively affect second-half performance due to decreases in body and muscle temperature [10], leading to reduced physical output [11,12]. The ability to perform high-intensity activities is essential in team sports [10]. Reductions in muscle temperature have been found to cause declines in high-intensity running in elite players, negatively impacting sprint and jump performance post-halftime [11,13]. Similarly, Russell et al. [14] and Lovell et al. [15] reported diminished physical performance and match intensity early in the second half due to physiological deactivation, characterized as the decline in physiological readiness during the halftime break, including reductions in muscle temperature, heart rate, and neuromuscular activation.

Active re-warm-up interventions can mitigate these effects [16], helping to preserve muscle temperature and neuromuscular function [4,17]. Studies suggest that protocols lasting 5 [15,18] to 7 min [11,19] improve second-half performance compared to passive rest [20], especially in the initial 15 min of the second half [11,15]. These re-warm-up interventions, often related to post-activation performance enhancement (PAPE), aim to optimize performance in high-intensity actions, such as jumping, sprinting, and changes of direction, by incorporating strength and power exercises [2,21].

Such findings highlight the importance of exploring halftime re-warm-up strategies that help maintain player readiness for soccer-specific high-intensity activities, including running and changes of direction [11,19,22]. However, despite the growing evidence supporting these interventions, a systematic review by Hammami et al. [10] emphasizes the lack of consensus on the most effective protocols. Variables such as activity type, intensity, and break duration appear to influence the effectiveness of these strategies, highlighting the need for further research to determine optimal re-warm-up approaches, particularly regarding PAPE-based methods and their impact on early second-half performance.

Several studies have documented the detrimental effects of passive halftime rest on performance, especially on sprinting and vertical jump height [23,24], primarily due to reduced muscle temperature, which impairs explosive force generation [11,24]. Re-warm-ups help preserve muscle temperature and neuromuscular function, thereby enhancing second-half performance [25]. Meta-analyses confirm the negative effects of passive halftime rest on vertical jump performance [25], while subjective assessments (e.g., ARMS) indicate reduced perceived muscular readiness [26]. These effects are most prominent in the first 15 min of the second half [14], reinforcing the necessity of effective halftime re-warm-up strategies to prevent performance declines in soccer [13]. Although re-warm-up strategies enhance second-half performance [1,13,27,28], implementation in professional environments is often hindered by time constraints and coaching preferences [1]. Therefore, it is recommended to explore high-intensity, short-duration (≈3 min) exercises [19]. Christaras et al. [24] found that a three-minute re-warm-up maintained core temperature and preserved high-intensity performance, specifically in jumping and sprinting in youth soccer players. These results suggest that short-duration, low-to-moderate intensity protocols (~3 min) may be practical and effective [19], especially when time is limited, to help maintain body temperature and improve sprint performance post-halftime, showing comparable effects to longer protocols [11,15,18].

Additionally, a major gap in the literature is the underrepresentation of female athletes in studies on warm-up and re-warm-up interventions [25]. The predominance of male-focused research is problematic given the rapid growth of women’s soccer [29]. Female athletes exhibit distinct physiological characteristics, such as muscle composition, thermoregulation, and hormonal profiles, that may influence their responses to re-warm-up strategies [30]. Recent studies [17,31,32] show that females respond differently to potentiation-based warm-up protocols, suggesting that male-derived findings may not be directly transferable. A systematic review by Silva et al. [17] emphasizes that most current knowledge is based on male cohorts, limiting applicability to female players. The lack of validated, female-specific protocols restricts the development of effective interventions. Recent reviews [33] emphasize the urgent need for sex-specific research, making this scientific gap a core motivation for the development and design of the present study.

This study aims to investigate the effects of two distinct halftime re-warm-up protocols on performance in a professional women’s soccer team: a strength-plyometrics-balance protocol (SPBP) and a soccer-specific protocol (SSP). The SPBP integrates neuromuscular exercises designed to induce (PAP), which enhances performance in activities requiring strength, speed, and power. The SSP, in contrast, focuses on soccer-specific movements, including small-sided games (SSG), and can be completed within a three-minute timeframe, making it a practical option for match settings. By focusing on female-specific responses, this research intends to fill the gap in the literature by providing evidence-based recommendations for re-warm-up protocols tailored to female athletes, thereby contributing to the development of more effective and practical strategies for improving second-half performance.

## 2. Materials and Methods

### 2.1. Study Design

A randomized trial with two parallel groups was conducted to compare the effects of two active re-warm-up protocols at halftime on physical performance and perception of readiness in women’s soccer players. The re-warm-up protocols (SPBP and SSP) were randomly assigned using simple randomization. However, the testing procedures followed a fixed order for all participants to ensure standardization and minimize variability. Following randomization, no relevant differences were observed between the groups. Research has demonstrated that the FIFA 11+ program can elicit acute improvements in performance, including a reduction in 20 m sprint time, an increase in jump height, and enhanced agility compared to other dynamic warm-ups [34]. The FIFA 11+ protocol was chosen for its simplicity and speed, with six exercises feasible within the three-minute re-warm-up period, aligning with practical competition settings. Consequently, we included a protocol comprising the following: (i) an SPBP integrating a segment of FIFA 11+ and (ii) an SSP featuring a small-sided game (i.e., 5 vs. 5 + Floater). The evaluation focused on performance in a 20-m sprint test and CMJ (Figure 1). The study’s objectives and procedures were explained to the players, who then signed an informed consent form. All procedures complied with the Declaration of Helsinki (updated version of 2013, Fortaleza).

### 2.2. Participants

We recruited a purposeful, convenient sample of 22 professional women’s soccer players of 23.4 ± 3.4 years; height: 168 ± 6.6 cm; body mass: 60.1 ± 6.0 kg (mean ± standard deviation), representing the entire roster of a club participating in the 2023/2024 Portuguese Women’s National Soccer Championship season. Prior to recruitment, approval was obtained from the team’s technical staff. Players who met the selection criteria were randomly assigned to the study groups. The study included female soccer players with no limitations on match participation. Exclusion criteria were as follows: (i) recent injury within the week prior to evaluations affecting match performance, (ii) players in the process of returning to match play, and (iii) individuals with pre-existing physical limitations before the evaluations.

The team coach, under the supervision of the study’s lead author, randomly assigned the 22 players into two groups using a balanced randomization method. Players were first paired based on specific characteristics, such as prior performance, and then randomly allocated within each pair, ensuring that both groups had a similar composition with respect to these key variables. This ensured a fair and unbiased assignment. Following randomization, each group consisted of 11 participants (Figure 2).

### 2.3. Interventions

#### 2.3.1. Familiarization Session

Before the start of the study, players participated in a familiarization session to become accustomed to the re-warm-up protocols and the tests designed to assess sprint performance and lower limb power. This session was conducted under conditions identical to those of the experimental session, including the use of GPS tracking to ensure consistency in data collection and monitoring. Players followed the exact same procedures, ensuring complete familiarity with all components of the protocol. The session began with a standardized warm-up identical to that performed before official matches. After the warm-up, players moved to the center of the field (with a maximum transition time of one minute to simulate halftime constraints) and performed a linear sprint, followed by an equivalent rest period by a CMJ. The first 45-min half of the simulated match then commenced.

During the final three minutes of the simulated halftime interval, players completed both re-warm-up protocols as instructed. At the end of this period, they returned to the center of the field (again within a one-minute transition window), performed the sprint test, followed by the CMJ, and then began the second 45-min half. At the end of the second half, while in the locker room, participants completed a post-exercise questionnaire. This familiarization process ensured that players were thoroughly prepared for each stage of the experimental protocol, reducing variability and enhancing the reliability of the performance and perceptual data collected.

#### 2.3.2. Data Collection Procedures

After completing the full familiarization session, participants returned on a separate day for the simulated first-half match, during which all procedures were reviewed again. In this initial phase, players performed their usual pre-match warm-up under the guidance of the technical staff, following the recommendations of the FIFA 11+ Program. This standardized warm-up lasts approximately 15 to 20 min and consists of three main parts:(1)running and dynamic mobility exercises to raise body temperature and activate the cardiovascular system;(2)strength, balance, plyometric, and neuromuscular control exercises aimed at improving motor control and reducing the risk of muscular and ligament injuries;(3)running with accelerations and changes of direction to prepare athletes for explosive actions during the match.

This protocol has been shown to improve balance, eccentric strength, and neuromuscular control and is recommended for athletes aged 14 years and older [34,35].

At the end of the warm-up, players performed the pre-test to assess sprint speed and lower limb power using the 20-m sprint test and CMJ, respectively. The tests were initiated immediately after the warm-up without any rest period in order to preserve the temporal effect of the warm-up, as per the established protocol. To ensure this, the investigative team and technical staff supported the process, and players were previously instructed to ensure an efficient and delay-free transition between the warm-up and the tests.

Next, participants played a simulated 11-a-side mach consisting of two 45-min halves and a 15-min halftime break, in accordance with the official rules of the Fédération Internationale de Football Association (FIFA). During the halftime break, athletes remained in passive rest in the locker room and were instructed not to engage in any physical activity that could affect the study’s outcomes. They were allowed to perform typical halftime activities such as using the restroom, listening to the coach’s instructions, and hydrating, but no additional physical exertion was permitted. Any deviation from these instructions was recorded and considered during data analysis.

In the last three minutes of the break, the SSP protocol was applied to 11 players, while the other 11 performed the SPBP, according to prior randomization. The post-test was then conducted to assess sprint speed and power, using the same procedures as the pre-test. Finally, the Acute Readiness Monitoring Scale (ARMS) questionnaire [36,37] was administered to evaluate players’ readiness and ability to perform tasks during the second half (Figure 1).

This structure is commonly used in soccer to replicate competitive match conditions and to allow for performance-related data collection under ecologically valid circumstances. Implementing the intervention in this context ensured that players were in a physical and mental state comparable to official competition while allowing for controlled application of the re-warm-up protocols. This approach has been adopted in previous studies, such as those by Edholm et al. [13], who conducted their interventions during competitive periods, and Yanaoka et al. [28], who simulated match-play environments to improve ecological validity.

#### 2.3.3. Strength, Plyometrics, and Balance Protocol (SPBP)

The SPBP protocol consisted of six exercises based on Part 2 of the FIFA 11+ Program, specifically designed to develop lower-limb strength, neuromuscular control, and dynamic balance and to reduce and improve the performance of soccer players through specific strength, balance, and running exercises [38]. The selected exercises were as follows: (i) squats with toe raise jumping, (ii) walking lunge squats, (iii) one-leg squats jumping, (iv) vertical jumps, (v) lateral jump squats, and (vi) box jumps. Using this re-warm-up protocol, players performed two sets of eight repetitions of each exercise three minutes before starting the aforementioned sprint and jump tests. The protocol was designed to fit within three minutes, as the six exercises are simple, expeditious, and do not require equipment. This enables efficient execution within the time constraints of a real competitive context. This approach aligns well with the practical constraints of halftime in soccer and enhances the ecological validity of the study.

#### 2.3.4. Soccer-Specific Protocol (SSP)

The SSP consisted of a small-sided game (5 vs. 5 + floater) aimed at maintaining ball possession on a 15 × 15 m field [39]. Players were randomly assigned to teams by the coaching staff, and the only rule was that each player was only allowed to touch the ball twice without other strategic focuses in order to reduce variables that could influence performance. The goalkeeper acted as the floater and was allowed to play with both feet and hands. The utilization of a floater is a common practice in small-sided games within the context of soccer, facilitating match dynamics. Researchers and technical staff positioned themselves along the sidelines with replacement balls to ensure continuous play and prevent interruptions when the ball went out of bounds. The intervention was continuous and lasted for three minutes, and the players performed both tests immediately after the re-warm-up, without any rest period prior to the test, to maintain the time window for the effects of the re-warm-up. All procedures of both protocols were strictly controlled by the research team.

### 2.4. Performance Assessment

After the warm-up and re-warm-up, players completed two physical tests to assess performance: a 20-m sprint and a CMJ test. Sprint performance was recorded using the WIMU PRO™ device from RealTrack Systems (Almería, Spain). The analysis was based on the average speed recorded during the 20-m sprint test. Each player was assigned an individual GPS device, which was positioned at midfield, activated 30 min before the initial warm-up, and placed in an adjustable vest worn on the player’s back. This GPS monitoring system reliably collects movement data during training and matches (ICC > 0.93) [40]. After the simulated match, the devices were removed, and the data were transferred to the proprietary S-PRO™ analysis software (Version 1, Hudl (formerly WIMU, RealTrackSystems), Lincoln, NE, USA).

To ensure accuracy, the start and end points of each 20-m sprint were identified using the GPS tracking data, and each sprint was individually isolated through the S-PRO™ software. This procedure enabled the precise calculation of the average speed over the exact 20-m distance for each athlete. The same methodology was applied at both evaluation time points (i.e., post-warm-up and post-re-warm-up), allowing for a direct and reliable comparison of performance across conditions for each player.

Following the sprint test, after a one-minute rest, players performed a CMJ to assess lower limb power. The CMJ height was measured using the My Jump 2 (MJ2) application (Carlos Balsalobre (the creator and proprietary of the software), Madrid, Spain), which has demonstrated nearly perfect agreement with force plate measurements (ICC = 0.997, 95% CI: 0.996–0.998). This application has been employed in other sports [41,42] and has shown nearly perfect agreement (ICC = 0.997, 95% CI: 0.996–0.998) compared to the gold standard method for estimating CMJ height (i.e., force plate) [43]. Players were instructed to stand upright with their hands on their hips and maintain this position throughout the test. When ready, they performed a downward movement until their knees reached a 90-degree angle, then immediately executed a rapid and explosive vertical jump to achieve maximum height. All jumps were recorded on video and later analyzed using the MJ2 app. Takeoff was defined as the moment both feet left the ground, and landing as the moment at least one foot touched down [40]. Each athlete performed one jump and repeated the movement if incorrectly executed. Only one sprint and one CMJ were performed to avoid compromising match readiness [44] and to maintain the time-sensitive blinding needed to assess the warm-up and re-warm-up effects on performance.

Although relying on a single sprint and jump trial may be viewed as a potential limitation, given that multiple attempts are often recommended to ensure maximal effort, all athletes in the present study underwent a familiarization session and were clearly informed in advance that only one attempt would be permitted. This likely enhanced their focus and execution. Considering the athletes’ professional status and the limited time available during the re-warm-up window, this approach is methodologically justified and consistent with previous research conducted in elite sport settings [45,46].

### 2.5. Secondary Outcomes

Following the physical tests, the players completed the Acute Readiness Monitoring Scale (ARMS) questionnaire [26], which consists of 32 items encompassing nine factors: general readiness, physical readiness, physical fatigue, cognitive readiness, cognitive fatigue, readiness for threat challenges, readiness for skills and training, group/team readiness, and equipment readiness [36]. To explore the implications of re-warm-up on players’ performance in the second half of the match, we selected items from the ARMS, which we considered most directly relevant to our goals: (i) physical readiness (three questions: 1, 14, and 16); and (ii) cognitive readiness (three questions: 26, 27, and 28) [36]. We acknowledge that this means the data are not validated, so we refrained from analyzing composite scores and instead examined each item separately, with the discussion of these results necessarily being cautious. The questionnaire utilized a scale from zero (did not apply) to six (fully applied) and was completed in the locker room. Players were instructed on completion and assured of response confidentiality. They were told not to communicate with teammates while filling out the questionnaire to ensure individual completion without external influence.

### 2.6. Statistical Analyses

The Kolmogorov–Smirnov test was used to analyze the data distribution, which followed the normal distribution. Therefore, means and standard deviations (SD) were reported. We used mixed ANOVA (within- and between-factors) to compare the group*time interaction for the primary outcomes (CMJ and 20-m sprint test). Multifactorial ANOVA was used to compare the responses of the ARMS questionnaire. The magnitude of the effect was assessed using partial eta squared (η_p_^2^*)* and classified as small (≥0.01), moderate (≥0.06), and large (d ≥ 0.14) [47]. The level of statistical significance was set at 5%. All analyses were performed in IBM SPSS software version 27.0 (Chicago, IL, USA).

## 3. Results

### 3.1. Main Outcomes: Interaction Between Protocols on Players’ Sprint and CMJ

The interaction between protocols showed no significant differences in sprint (*p* = 0.68, η_p_^2^ = 0.008) or CMJ (*p* = 0.07, η_p_^2^ = 0.004) performances (see Table 1). Players were slower at post-re-warm-up compared to post-warm-up, with an effect of large magnitude (*p* < 0.001, η_p_^2^ = 0.79).

There was no significant difference in CMJ (*p* = 0.07, η_p_^2^ = 0.151) between post-warm-up and post-re-warm-up. In the inter-group analysis, no significant differences were observed between the protocols in sprint (*p* = 0.88, η_p_^2^ = 0.001) or in CMJ (*p* = 0.06, η_p_^2^ = 0.162), indicating that both rewarming methods produced similar responses (Figure 3).

### 3.2. Secondary Outcomes

There were no significant differences in responses to the ARMS questionnaire (*p* = 0.481, η_p_^2^ = 0.27) between players who performed the SSP and those who completed the SPBP protocol (Table 2).

## 4. Discussion

Our aim was to investigate and compare the effects of two active re-warm-up protocols (SSP and SPBP), applied during the last three minutes of halftime, on players’ physical performance, specifically sprint and CMJ, measured at two time moments: post-warm-up and post-re-warm-up. Both protocols were designed to be feasible within the short halftime re-warm-up window and resulted in no significant interaction effects on sprint or CMJ performance, nor on perceived readiness as assessed by the ARMS questionnaire.

The underlying mechanisms of the SSP and SPBP protocols provide valuable insights into their effects on performance enhancement. A detailed analysis of these mechanisms, as highlighted in the review by McGowan et al. [4], reveals that strength-related and plyometric exercises primarily promote benefits through post-activation potentiation (PAP) or post-activation performance enhancement (PAPE), whereas small-sided games (SSGs) mainly increase muscle temperature and neuromuscular readiness. This distinction offers a clearer understanding of how each re-warm-up protocol may differentially influence athlete performance. Our findings can be contextualized by comparing the characteristics of re-warm-up interventions used in previous studies. Variations in re-warm-up duration, intensity, and exercise modality (e.g., cycling, running, resistance exercises) can explain discrepancies in outcomes across the literature.

The temporal pattern of match intensity deserves particular attention, as the first 15 min of the second half represent a unique period often characterized by lower physical output despite the accumulated fatigue expected in the later stages of the match. Evidence from Mohr et al. [11], Russell et al. [14], and Lovell et al. [15] demonstrates that reductions in high-intensity running and sprint performance during this period are likely related to physiological factors such as decreased muscle temperature and neuromuscular deactivation during the passive halftime interval. Re-warm-up protocols can mitigate these declines by maintaining muscle temperature and neuromuscular readiness, as supported by Silva et al. [2] and McGowan et al. [4], which demonstrated the benefits of re-warm-up protocols in minimizing these effects focused specifically on physical performance.

In this context, the present investigation contributes to the development of evidence in this field by highlighting the importance of re-warm-up strategies in mitigating the performance declines typically observed after halftime. It offers a deeper understanding of their role in optimizing match intensity during the early stages of the second half, particularly in relation to sprinting, jumping actions, and attentional focus on task execution, assessed through the application of the ARMS questionnaire. These considerations strengthen the interpretation of our results and better situate our findings within the existing literature on re-warm-up effects and match performance dynamics.

The discussion was structured to emphasize the comparative analysis between the two re-warm-up protocols, aligning with the study’s primary objective, rather than providing an exhaustive examination of each protocol in isolation. The data showed that sprint times significantly increased (indicating a decline in performance) after the re-warm-up in both protocols (SSP: +0.35 s; SPBP: +0.38 s; *p* = 0.001), with a large effect size. CMJ height decreased slightly (SSP: −1.2 cm, *p* = 0.074; SPBP: −1.6 cm, *p* = 0.074), suggesting small magnitude effects without statistical significance. These findings align with some previous studies [13], although other research has reported improved jumping and sprinting performance following re-warm-ups [20,28,48], highlighting the variability depending on protocol design and context.

Although sprint performance declined after the re-warm-up protocol, it is unlikely that protocol-specific fatigue was the cause. The protocols involved short-duration activities (3 min) at moderate to low intensities, which are generally designed to maintain or enhance neuromuscular readiness rather than induce fatigue. It is possible that for some participants, particularly those less accustomed to such protocols, the re-warm-up may have contributed to transient fatigue and, consequently, affected sprint performance, but we consider this to be unlikely. Still, future studies could incorporate a subjective rating of perceived exertion (RPE) scale to assess the impact of the protocol on fatigue induction in athletes. Similar findings have been reported by other authors; for example, Hammami et al. [10] and Edholm et al. [13] also observed reductions in sprint performance following certain re-warm-up strategies, suggesting that suboptimal intensity, accumulated fatigue, or insufficient recovery time could attenuate the expected benefits of re-warm-up protocols. It is conceivable that the accumulated fatigue from the first half of the match simulation may have contributed to the decline in sprint performance, regardless of the re-warm-up intervention, as fatigue has been shown to impair neuromuscular function and sprint capacity during match-play [17].

Given this context, our primary objective was not solely to determine whether sprint performance improved after re-warm-up but rather to compare the magnitude of performance decline between protocols. By examining which protocol better attenuated the performance drop both in sprint and jump performance, we aimed to identify the most effective strategy to preserve physical output during halftime. Moreover, the lack of significant differences in jump performance (CMJ) post-intervention suggests that fatigue effects, if present, were limited in scope and did not broadly impair explosive lower-limb power. These findings underscore the importance of balancing the intensity and duration of re-warm-up protocols to avoid inducing additional fatigue while preserving performance benefits, but fatigue accumulated during the first half of the match will still likely play a role. Future research could incorporate measures of fatigue, such as subjective ratings and biochemical or neuromuscular markers, to better understand its potential confounding role in performance outcomes.

The similar effects observed between the SSP and SPBP protocols reinforce the practical implication that short, feasible re-warm-up routines can be integrated into soccer halftimes despite time and logistical constraints. Both protocols demonstrated comparable impact on performance and perceived readiness; we prioritized presenting these overall comparative results to maintain a clear and focused narrative aligned with the study’s primary aims. The inclusion of a control group with passive rest could have strengthened the analysis of the absolute effectiveness of each protocol; however, the primary objective of this study was to compare two re-warm-up protocols that are already supported by previous literature for their ability to mitigate performance decline after halftime, although they operate through distinct mechanisms.

The SPBP protocol is grounded in neuromuscular and proprioceptive stimulation principles [34,49,50], while the SSP protocol is based on the contextual specificity of match demands [22,39,51]. Therefore, our focus was to investigate which of these two approaches would prove more effective within the practical constraints of a competitive halftime period, particularly considering the very limited time available (≤3 min). We chose not to include a passive control group, as several studies have consistently shown the detrimental effects of passive rest on both physical and technical-tactical performance at the beginning of the second half [10,11,52]. This allowed us to concentrate our efforts on directly comparing the two active re-warm-up protocols under ecologically valid conditions.

The environmental factors, such as ambient temperature, humidity, and playing surface conditions, could potentially influence the magnitude of body temperature fluctuations during the halftime interval and subsequently affect performance parameters in the second half. Previous research has demonstrated that environmental conditions can significantly impact thermoregulatory responses in athletes [53,54], potentially modulating the effectiveness of re-warm-up interventions. Future studies should consider incorporating systematic measurement of environmental variables and potentially explore how different re-warm-up protocols might be optimized for specific environmental conditions, particularly in women’s soccer, where research on environmental influences on performance remains limited.

Both re-warm-up protocols (SSP and SPBP) were feasible considering the short three-minute time window, and there were no significant interaction effects regarding sprint or CMJ performance or between-group differences in perceived readiness to perform. Comparing pre-match to post-re-warm-up values for both protocols (SSP and SPBP), significant increases were observed in sprint times, with variations of +0.35 and +0.38 (*p* = 0.001), respectively, indicating a large magnitude effect for both. Similar effects were noted for the CMJ; however, these did not reach statistical significance (SSP: −1.2 cm, *p* = 0.074; SPBP: −1.6 cm, *p* = 0.074), suggesting a small magnitude effect for both.

It was shown previously that re-warm-up strategies could mitigate the negative impact of passive halftime practices on physiological measures (heart rate and core temperature) and on players’ performance (jump, sprint, and distance covered) [10]. However, the available time for a re-warm-up in soccer is estimated to be under three minutes [1], so longer protocols may not be as easily implemented in real-world contexts [55]. The SSP and SPBP protocols were designed to be implemented within the available re-warm-up time window (approximately three minutes). In the case of SPBP, the selection of only six exercises from FIFA 11+ was strategically made to ensure its feasibility and practical application in a soccer context. The simplicity of both protocols facilitated their efficient execution within the limited time frame. This suggests that re-warm-up protocols can be effectively implemented in practical contexts, even when time is constrained.

Although the intervention was carried out in a simulated match environment, the protocols were carefully designed to replicate the constraints and demands of an official soccer match, thereby enhancing ecological validity and facilitating the applicability of findings to real-world settings. Utilizing experimental conditions that closely mirror actual game scenarios is essential to ensure that research outcomes are relevant and transferable to practical performance contexts [5]. Despite the temporal limitations imposed by soccer regulations on re-warm-up during halftime, our findings indicate that short and feasible re-warm-up protocols can be implemented within a 3-min halftime window. However, given that sprint performance declined compared to the post-warm-up condition, it is not possible to assert positive effects on performance. Rather, the implementation of such protocols may help attenuate performance losses typically observed after passive halftime periods.

Considering that current regulations restrict access to the field before the start of the second half in official matches and that the space available outside the playing field is often inadequate, this situation could pose a challenge to the implementation of the re-warm-up protocols.

However, the adaptation of the SSP protocol exercises, based on the FIFA 11+ Program, proved to be highly feasible without compromising their execution and ensuring their viability in real scenarios. The findings encourage potential changes to match regulations, considering the positive effects of SSP on performance. The two protocols had similar effects on sprint and CMJ performance, with no significant interaction effects. The average SSP performance decreased from 32.3 cm to 31.1 cm (mean difference: −1.2 cm; *p* = 0.074), while the SPBP performance decreased from 36.3 cm to 34.7 cm (mean difference: −1.6 cm; *p* = 0.074). This corresponds to a percentage decline in CMJ performance of 3.72% for SSP and 4.41% for SPBP following the re-warm-up.

This suggests that CMJ performance slightly decreased after the re-warm-up in both protocols, although differences did not reach statistical significance, with a slightly larger decrease in the SPBP protocol compared to the SSP protocol. This decline aligns with findings by Edholm et al. [13], who reported a decrease of 7.6% in the control group and 3.1% in the re-warm-up group. However, research has also shown improved jumping performance after a re-warm-up [56].

These discrepancies in the literature can be attributed to various methodological factors. Variations in duration, exercise type, and intensity of re-warm-up protocols appear to be determinant factors for the observed outcomes. For instance, while Yanaoka et al. [19] demonstrated benefits with short-duration re-warm-up protocols (3 min) at both moderate intensity (60% VO_2_max) and low intensity (30% VO_2_max), Hammami et al. [10] reviewed studies employing re-warm-up protocols lasting between 5 and 7 min at varying intensities, highlighting their significant impact on maintaining or enhancing physical performance after the halftime break. Similarly, Mohr et al. [11] found that a 7-min moderate-intensity re-warm-up improved sprint performance by 2.8% compared to passive rest. Additionally, participant characteristics, such as fitness level, sport-specific training background, and physiological adaptations, may contribute to variable responses to re-warm-up protocols, as highlighted in a recent systematic review by Zois et al. [39]. For instance, Sanchez-Sanchez et al. [44] demonstrated that using elastic bands during half-squats as a re-warm-up improved strength, power, and speed in youth soccer players.

Other factors explaining conflicting results include the diverse dependent variables and assessment methods employed across studies. While some researchers focused on specific physical performance measures (single vs. repeated sprints, vertical jump, agility), others also included physiological parameters such as muscle temperature or metabolic responses. Russell et al. [55] demonstrated that a 2-min re-warm-up attenuated the decline in core temperature by 0.5 °C and improved repeated sprint ability immediately after halftime in professional rugby players. This finding suggests that short re-warm-up protocols can be effective in maintaining muscle temperature and enhancing physical performance during rest periods in sports. Whereas Lovell et al. [15] found improvements primarily in muscle glycogen resynthesis rather than performance metrics. Environmental conditions (temperature, humidity) and the precise timing of re-warm-up during the interval (beginning, middle, or end) also significantly affect protocol efficacy and contribute to the variability of results reported in the literature. A recent meta-analysis [57] identified timing as a critical factor, with re-warm-ups performed within the final 3 min of halftime showing greater efficacy than those performed earlier. This complexity underscores the importance of considering multiple factors when interpreting and comparing results from different studies on re-warm-up protocols.

Our results indicated a significant decline in sprint performance following the re-warm-up protocols. Specifically, sprint times were longer after the re-warm-up compared with post-warm-up (SSP: 3.86 ± 0.15 s vs. 3.51 ± 0.15 s; SPBP: 3.88 ± 0.26 s vs. 3.50 ± 0.10 s). Conversely, Yanaoka et al. [19] investigated two distinct 3-min re-warm-up protocols, rather than a single protocol with variable durations. Specifically, these protocols consisted of the following: (1) 3 min of cycling at 60% of VO_2_max (moderate intensity) and (2) 3 min of cycling at 30% of VO_2_max (low intensity). Both protocols were compared to the control condition (passive rest) and reported improvements in sprint performance following re-warm-up protocols involving either one minute of cycling at 90% VO_2_max or three minutes at 30% VO_2_max. Notably, in their study, sprint performance following the re-warm-up was superior to that observed after passive rest during halftime. These differences may reflect variations in the re-warm-up protocols used and participant characteristics.

Furthermore, Ltifi et al. [20] investigated the impact of brief re-warm-up activities on the sprint performance of youth soccer players. The intervention consisted of a re-warm-up protocol lasting only 3 min, during which players wore weighted vests corresponding to 5% and 10% of their body mass. The results demonstrated that this re-warm-up approach, using weighted vests, significantly improved 20-m sprint performance. Performance measurements were compared between two time points: after the initial warm-up and after the re-warm-up intervention, allowing for a direct assessment of the effect of this “micro-dose” re-warm-up on subsequent performance. The study concluded that three-minute re-warm-up with a vest weighing 10% of body mass yielded the highest RPE and notable improvements in 20-m sprint performance, suggesting that young elite soccer players should incorporate 10% body mass vests into their re-warm-up routines to enhance sprint performance after halftime.

Overall, these studies highlight the importance of effective re-warm-up strategies in maintaining and improving sprint performance in soccer. By optimizing these protocols, coaches and trainers can better prepare players for the demands of the match, particularly during critical moments such as the start of the second half. Our results therefore suggest that both protocols produced similar outcomes on sprint and CMJ performance and perceived readiness, which directly addresses the main research question; however, no passive control group was available to better frame the relevance of these findings. From a perceptual standpoint, the two protocols (SSP and SPBP) resulted in similar perceived readiness as assessed through six questions from the ARMS questionnaire. It is important to consider that the duration of the re-warm-up may have influenced our results.

While a 3-min protocol is more feasible in real competitive contexts, the literature suggests that longer periods of 5 to 8 min may be necessary to achieve an adequate re-warm-up [2]. Future research should investigate the optimal balance between the physiological benefits and the practical feasibility of re-warm-up strategies, particularly in light of the limited halftime period, often constrained by coaching practices as noted by Towlson et al. (2013) [1]. Moreover, studies should consider additional contextual factors aligned with the specific needs of each team, including competition level, environmental conditions, and individual athlete profiles, to ensure the development of tailored and effective interventions. The protocols applied in the present study—both the SPBP and the SSP—may serve as foundational models for the design of practical re-warm-up strategies adapted to various sporting contexts.

### Limitations

The small sample size, which also resulted in the absence of a passive control group, limits the ability to assess the effectiveness of the two proposed re-warm-up protocols in comparison to rest. However, as demonstrated by the literature, a significant decline in physical performance occurs following a passive halftime period [12,16,27], with documented reductions in muscle temperature, sprint capacity, and lower-limb power [14], which reinforces the importance of re-warm-up strategies to mitigate these negative effects and optimize subsequent performance.

The variation in methodologies across studies prevents more direct comparisons, and it is possible that dose-response relationships mediate the results, which should be explored in future research. While a comparison with a control group would have provided a clearer reference point to isolate the specific effects of the re-warm-up interventions, the study design still offers meaningful insights into the relative effectiveness of the two protocols.

To preserve the ecological validity of the study, we deliberately chose not to assess performance at the end of the first half of the simulated match. In real-world scenarios, coaches typically use this moment to communicate with players, and conducting performance tests at this time could interfere with this essential process. Additionally, any performance decline observed after halftime could be largely attributed to accumulated fatigue during the match [17,58], without providing insights into the effectiveness of the re-warm-up protocols.

Beyond the limitations previously identified (i.e., passive control groups and larger samples), future research should consider incorporating additional performance metrics, as relying solely on sprint and CMJ tests provides a limited view of overall athletic performance. Although all participants underwent a familiarization session and were professional athletes, the use of a single trial for sprint and CMJ assessments may reduce measurement reliability. Multiple attempts are typically recommended in similar studies to better capture maximal performance and minimize variability.

Moreover, there is a need to develop instruments capable of assessing perceived readiness to perform that are straightforward and can be applied quickly. Practical constraints inhibited the implementation of the crossover design. Specifically, it would require greater interference with the team’s natural training process. Therefore, we decided not to use a crossover design, despite the limitations of doing so.

## 5. Conclusions

This study provides insights into the effects of a short-duration re-warm-up protocol (3 min) on the physical performance of soccer players. While this duration may be more feasible in real competitive contexts, our results indicate that it may not be entirely effective in restoring all aspects of physical performance, particularly sprint times. These findings highlight the need to balance practical considerations with physiological requirements when designing re-warm-up protocols for sports competitions. Both re-warm-up protocols (SSP and SPBP) produced similar effects, specifically increases in sprint times and non-significant reductions in jump performance compared to post-warm-up. The lack of a control group precludes us from stating whether the investigated protocols could mitigate the performance decrement related to fatigue accumulation. Nonetheless, our results show that SSP and SPBP did not have differential effects on the readiness of players to start the second half of the match. Coaches and fitness trainers should consider these findings when planning re-warming sessions, adjusting protocols as necessary to balance potential benefits and adverse effects on athletic performance. However, the implementation of re-warm-up protocols in official matches will require regulatory changes, particularly for those involving tactical and technical actions on the field, which remains a key consideration for future application.

## Figures and Tables

**Figure 1 jfmk-10-00270-f001:**
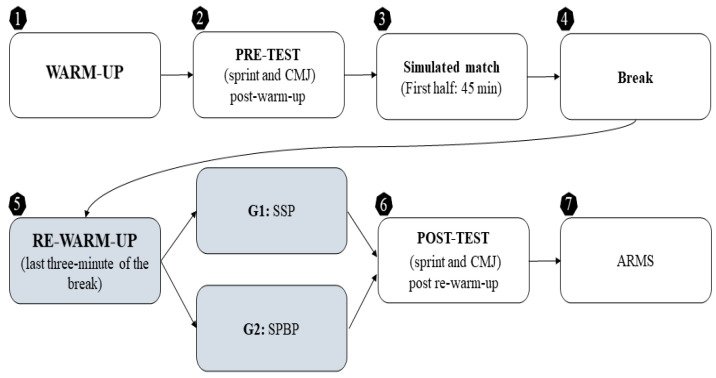
Diagrammatic representation of the randomized trial. Note: CMJ, countermovement jump; min, minutes; G1, Group 1; G2, Group 2; SSP, soccer-specific protocol; SPBP, strength, plyometrics, and balance protocol; ARMS, acute readiness monitoring scale.

**Figure 2 jfmk-10-00270-f002:**
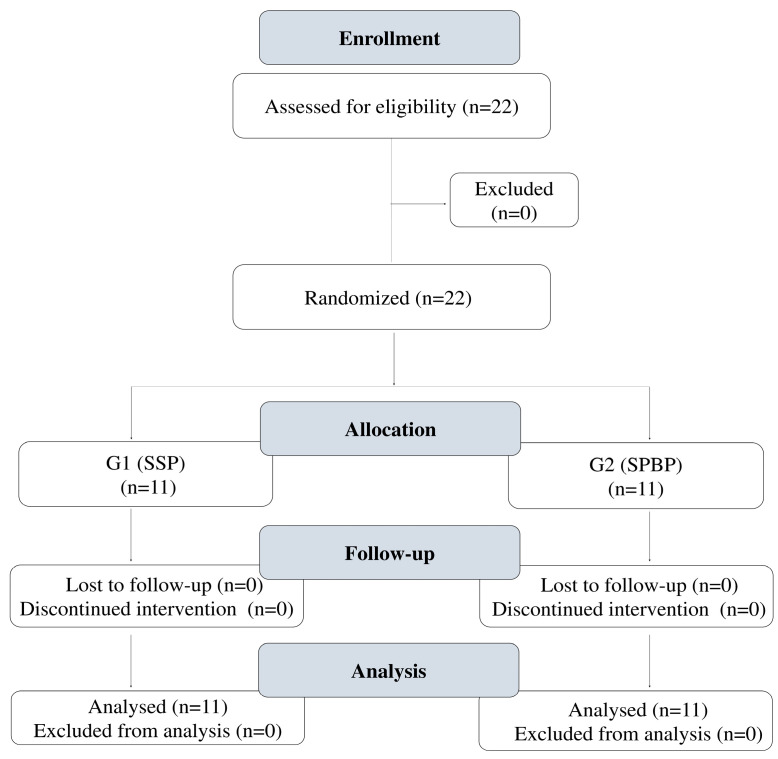
Flowchart of study participants. Note. SSP, soccer-specific protocol; SPBP, strength, plyometrics, and balance protocol.

**Figure 3 jfmk-10-00270-f003:**
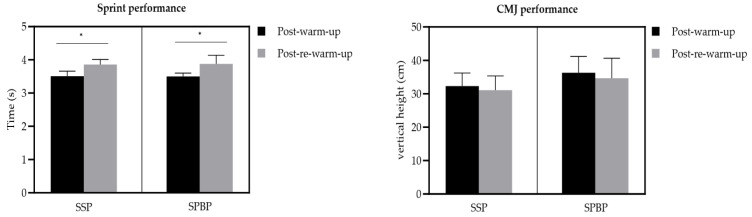
Effects of different protocols on sprint and countermovement jump performance. Sprint performance (left panel): Sprint times (in seconds) post-warm-up and post-re-warm-up for two protocols: an SSP, soccer-specific protocol, and SPBP, a strength, plyometrics, and balance protocol. ***** Significant difference (*p* < 0.001). CMJ, Countermovement jump performance (right panel): jump height (in centimeters) measured post-warm-up and post-re-warm-up under the same protocols. Data are presented as mean ± SD.

**Table 1 jfmk-10-00270-t001:** Descriptive statistics and interaction effects for sprint and CMJ, post-warm-up and post-re-warm-up.

Protocol		Mean ± SD	*F*	*p*	η_p_ ^2^
	**Sprint**		73.1	0.001 *	0.790
	**Sprint x Protocol**		0.17	0.684	0.008
**SSP**	Pre-test: Sprint post-warm-up (s)	3.51 ± 0.15			
	Post-test: Sprint post-re-warm-up (s)	3.86 ± 0.15			
**SPBP**	Pre-test: Sprint post-warm-up (s)	3.50 ± 0.10			
	Post-test: Sprint post-re-warm-up (s)	3.88 ± 0.26			
	**CMJ**		3.54	0.074	0.151
	**CMJ x Protocol**		0.85	0.773	0.004
**SSP**	Pre-test: CMJ post-warm-up (cm)	32.3 ± 3.93			
	Post-test: CMJ post re-warm-up (cm)	31.1 ± 4.27			
**SPBP**	Pre-test: CMJ post-warm-up (cm)	36.3 ± 4.88			
	Post-test: CMJ post re-warm-up (cm)	34.7 ± 5.94			

**Note.** * Statistically significant (*p* < 0.05). Abbreviations: CMJ, counter-movement jump; SD, standard deviation; SSP, soccer-specific protocol; SPBP, strength, plyometrics, and balance protocol.

**Table 2 jfmk-10-00270-t002:** Acute readiness monitoring scale questionnaire.

ARMS	SSP	SPBP	*p*
**Physical Readiness**			
I am physically fit	4.27 ± 1.27	4.72 ± 1.19	0.397
I am physically prepared	4.63 ± 1.36	4.63 ± 1.20	1.000
I am physically fresh	3.63 ± 1.28	3.90 ± 1.22	0.616
**Cognitive Readiness**			
I can focus well	4.63 ± 0.50	5.09 ± 0.94	0.174
I am mentally prepared	4.36 ± 1.36	5.09 ± 1.22	0.202
I am thinking clearly	5.00 ± 1.09	5.00 ± 0.89	1.000

Abbreviation: SSP, soccer-specific protocol; SPBP, strength, plyometrics, and balance protocol; ARMS, Acute readiness monitoring scale.

## Data Availability

The data that support the findings of this study are available from Marco Abreu (profmarcoabreu@hotmail.com) upon request, in compliance with policies regarding data sharing at the FADEUP, Porto, Portugal.

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
