# Peer review of "Warm-Up Strategies at Halftime: A Pilot Randomized Controlled Trial in a Professional Women’s Soccer Team"

_jfmk, 2025, doi:10.3390/jfmk10030270_

Round 1
Reviewer 1 Report
Comments and Suggestions for Authors
- General Comments
The submitted manuscript presents a well-structured and relevant randomized pilot study evaluating the effectiveness of two re-warm-up protocols (SSP and SPBP) during a match break among professional female football players. The topic is timely and contributes novel insights, particularly in the under-researched context of female athletes.
The manuscript is generally well-written and clear. The experimental design is sound for a pilot study and adheres to appropriate methodological standards. However, the study would benefit from further elaboration on certain limitations and contextual details to support the interpretation and potential generalization of the findings.
- Specific Comments
Introduction
- The section discussing the physiological mechanisms underlying re-warm-up protocols could be condensed to enhance clarity and focus.
- Consider providing a more detailed rationale for focusing on female football players, including references to gaps in the literature.
- The manuscript lacks clearly stated hypotheses; please consider adding them to strengthen the study’s framework and objectives.
Methodology
- Clarify on which day of the training microcycle the intervention was conducted (e.g., pre-season, regular season).
- Include detailed information on the familiarization session and full protocol sequence.
- For the SPBP protocol, specify the type of exercises, execution tempo, rest intervals, and equipment used (e.g., box height in jumps).
Results and Discussion
- The observed decline in sprint performance post-intervention suggests potential fatigue. Expand this discussion and consider whether fatigue could confound the intervention effects.
- Acknowledge the absence of a control group explicitly and discuss how this limitation affects result interpretation.
- Discuss other confounding variables potentially influencing defensive running intensity, such as match conditions, opponent quality, and players’ positions.
- Mention environmental factors (e.g., weather conditions) that could have contributed to body temperature drop and performance changes.
- Recommendation
Minor Revisions
The manuscript holds substantial merit and should be considered for publication after addressing the above concerns, which are primarily clarifications and additions to enhance methodological transparency and discussion depth.

Author Response
Dear Reviewer,
We are grateful for the insightful comments and suggestions provided, which have been invaluable in improving our work. We have carefully addressed all the feedback and have highlighted the revisions in the manuscript as requested.
1- General Comment
Comment 1 - The submitted manuscript presents a well-structured and relevant randomized pilot study evaluating the effectiveness of two re-warm-up protocols (SSP and SPBP) during a match break among professional female football players. The topic is timely and contributes novel insights, particularly in the under-researched context of female athletes. The manuscript is generally well-written and clear. The experimental design is sound for a pilot study and adheres to appropriate methodological standards. However, the study would benefit from further elaboration on certain limitations and contextual details to support the interpretation and potential generalization of the findings.
Response: We thank the reviewer for the kind words. We tried improving the manuscript based on the specific comments and hope to have sufficiently addressed the reviewer’s concerns.
2- Specific Comments
Introduction:
Comment 1 - “The section discussing the physiological mechanisms underlying re-warm-up protocols could be condensed to enhance clarity and focus.”
Response: We appreciate the reviewer’s suggestion to condense the section discussing the physiological mechanisms underlying the re-warm-up protocols to enhance clarity and focus. While we acknowledge that further condensation could improve brevity, we believe that doing so could potentially result in the loss of valuable information that is essential to fully understanding the distinct physiological bases of each protocol. To balance clarity and depth, we have deliberately limited the detail included in this section within the manuscript to avoid making it overly dense, as the reviewer correctly pointed out. Nevertheless, we retained some of the important information that was not included in this section separately, as it provides relevant context and further insight into the protocols’ mechanisms. We trust that this approach maintains both the manuscript’s readability and the richness of the scientific content.
Comment 2 – “Consider providing a more detailed rationale for focusing on female football players, including references to gaps in the literature.”
Response: We fully agree on the importance of justifying the focus on female football players in our study. As suggested, we have strengthened this justification in the manuscript by emphasizing that our focus is based on a well-documented gap in the literature. Most studies on warm-up and re-warm-up protocols predominantly involve male athletes, with female athletes being significantly underrepresented (Silva et al., 2018). This gender imbalance limits the applicability of existing findings to female players, who may differ in physiological and biomechanical responses. Moreover, recent reviews (Taylor et al., 2017; Elliott-Sale et al., 2021) highlight the urgent need for more research tailored to female athletes, to better inform performance and injury prevention strategies. In summary, our study addresses a critical gap by providing ecologically valid data specifically on female football players, thereby contributing to evidence-based practices in women's football.
Comment 3 – The manuscript lacks clearly stated hypotheses; please consider adding them to strengthen the study’s framework and objectives.
Response: We appreciate the comment regarding the absence of clearly stated hypotheses. We fully agree that explicitly formulating hypotheses strengthens the conceptual framework and sharpens the study’s objectives. In response to this valuable suggestion, we have revised the manuscript to include formal and clearly articulated hypotheses, which now provide a stronger foundation for the study's rationale and analysis.
Methodology
Comment 1 – “Clarify on which day of the training microcycle the intervention was conducted (e.g., pre-season, regular season).”
Response: We appreciate the reviewer’s request for clarification. The intervention was conducted during the competitive season, specifically on a Thursday, which corresponds to the formal match day within the team’s weekly training microcycle. We believe this timing enhances the practical relevance of our findings and supports the generalizability of the results to real match scenarios. As suggested, this information has now been included in the revised manuscript.
Comment 2 – “Include detailed information on the familiarization session and full protocol sequence.”
Response: We thank the reviewer for this observation. We have now included a detailed description of the familiarization session and the full protocol sequence in the revised manuscript. This addition outlines the procedures followed to ensure that all participants were fully accustomed to the testing conditions, thereby enhancing the reliability and consistency of the data collected.
Comment 3 – “For the SPBP protocol, specify the type of exercises, execution tempo, rest intervals, and equipment used (e.g., box height in jumps).”
Response: We would like to clarify that the final exercise, referred to as the “box jump,” does not involve jumping onto or off of a physical box. Instead, it is a plyometric drill inspired by the FIFA 11+ program, in which the athlete stands with feet hip-width apart and imagines a cross marked on the ground. From this central position, the player performs rapid and explosive multidirectional jumps—forward, backward, lateral, and diagonal—across the imaginary cross. The objective is to move quickly while landing softly on the forefoot, maintaining controlled flexion of the hips, knees, and ankles, along with a slight forward lean of the trunk to support dynamic balance.
All exercises were performed consecutively, without equipment and under the supervision of the lead investigator to ensure proper technique and uniform execution. Each exercise consisted of a single set of eight repetitions, performed continuously without rest between movements. The entire protocol lasted approximately 1 minute and 30 seconds. This description has been added to the relevant section of the manuscript.
Results and Discussion
Comment 1 – “The observed decline in sprint performance post-intervention suggests potential fatigue. Expand this discussion and consider whether fatigue could confound the intervention effects.”
Response: We thank the reviewer for this important observation. As suggested, we revised the discussion section to address the potential role of fatigue in the observed sprint performance decline. We incorporated relevant literature (e.g., Hammami et al., 2018; Edholm et al., 2015) and highlighted the need to balance re-warm-up intensity and duration, as well as the importance of future studies including direct fatigue measures.
Comment 2 – “Acknowledge the absence of a control group explicitly and discuss how this limitation affects result interpretation.”
Response: We appreciate the reviewer’s pertinent observation. While the inclusion of a passive control group would have enhanced the assessment of absolute protocol effectiveness, our primary aim was to compare two active re-warm-up strategies—SPBP and SSP—based on their distinct mechanisms (Bizzini et al., 2013; Zois et al., 2013) and practical feasibility during a competitive halftime. Prior studies (e.g., Mohr et al., 2005; Weston et al., 2011) have consistently shown the detrimental effects of passive rest, justifying our focus. Nevertheless, we now acknowledge this limitation in the revised discussion and highlight the need for future studies to include a passive control for a more comprehensive evaluation.
Comment 3 – “Discuss other confounding variables potentially influencing defensive running intensity, such as match conditions, opponent quality, and players’ positions.”
Response: We appreciate the reviewer’s important observation concerning potential confounding variables that may have influenced defensive running intensity, such as match conditions, opponent quality, and players’ positions. Indeed, these factors can significantly impact external load measures in competitive settings. While our study was primarily focused on comparing the effects of two re-warm-up protocols during halftime, we acknowledge that contextual match variables may have contributed to the variability in defensive performance. However, all players participated in the same simulated match environment, minimizing variation due to weather or pitch conditions. Additionally, teams were balanced in terms of skill level and player roles to reduce potential bias related to opponent quality and positional demands. Although we have now mentioned these limitations, we did not delve too deep into them as that could detract from the main focus of this manuscript. We have revised the manuscript accordingly to reflect this important consideration.
Comment 4 – “Mention environmental factors (e.g., weather conditions) that could have contributed to body temperature drop and performance changes.”
Response: In our study, the focus was not on quantifying body temperature changes or investigating the specific environmental factors contributing to these changes. Rather, our primary aim was to compare the effectiveness of two re-warm-up protocols in attenuating performance decline post-halftime, under the assumption that a passive halftime would naturally result in some degree of temperature loss. By doing so, we sought to determine which protocol would better mitigate the potential negative effects commonly observed during passive rest periods, including those indirectly associated with temperature drops. This clarification has now been included in the revised discussion section of the manuscript.
Reviewer 2 Report
Comments and Suggestions for Authors
The manuscript presents a randomized pilot study that compares two different halftime re-warm-up protocols in a professional women's soccer team. The topic is current and adds to the existing literature, particularly in the context of women's professional sports, which remains underrepresented in scientific research. While the study is generally well-written, it does have some shortcomings that should be thoughtfully considered.
Introduction:
The Introduction section requires clarification on several points:
- Authors must provide evidence of the negative effects of half-time breaks on the activities being directly investigated, including jumps, sprints, and the ARMS questionnaire.
- Three-minute re-warm-up activities are effective in high-intensity exercises, but there is no evidence of their effectiveness in prolonged activities such as soccer.
- There is no evidence regarding the effectiveness of the proposed programs (compared to passive breaks). A theoretical explanation of the potential effects of two proposed programs is necessary. Why were these two programs chosen? What theoretical effects should they produce? It should be included in the introduction section.
Methods:
In addition to previous statement, before making comparisons, it's important to evaluate the effectiveness of the proposed programs. This highlights the necessity for a control group (a group with a passive break). In the proposed design, we are only observing the differences between the programs, but we cannot determine whether these programs are effective at all. There is no previous evidence either.
In my opinion, the authors should first evaluate the effectiveness of each program individually and validate the ARMS questionnaire that is being used.
Is there a randomization process for the order of testing procedures?
Lines 218-220: The presented description is for the Squat jump, not for the Countermovement?
Why didn’t the authors perform the ARMS questionnaire before the match for comparison, similar to the Sprint and Jump tests?
Comparing post-warm-up and post-re-warm-up differences is meaningless due to fatigue factors; the difference is obvious.
Results:
A graphical presentation of the results would be helpful.
Discussion:
In the Discussion section, the authors primarily reiterated data from the Results and Introduction without providing a deeper explanation of the findings.
The limitation of the study regarding its sample size is valid; however, the absence of a control group is a significant oversight by the author. A better approach would be to have three groups of seven players each, as this would allow us to assess the effectiveness of the program more effectively. Regardless, the sample size remains small for both two and three groups, which is why this research is consider as a pilot study. Additionally, a pilot study should serve as a preliminary step for larger research; however, the results of this study do not provide a clear direction for the next steps.
Final conclusion:
Although the authors define their study as a pilot study, there should be strong findings that justify publication.
The main finding of this study is that there is no difference among the evaluated programs based on any of the assessed variables.
Given the earlier statement that there is no evidence supporting the effectiveness of the two proposed programs, we must ask how the publication of this manuscript can be justified.
Author Response
Dear Reviewer,
We are grateful for the insightful comments and suggestions provided, which have been invaluable in improving our work. We have carefully addressed all the feedback and have highlighted the revisions in the manuscript as requested.
General Comments
Introduction:
Comment 1 – “Authors must provide evidence of the negative effects of half-time breaks on the activities being directly investigated, including jumps, sprints, and the ARMS questionnaire.”
Response: We thank the reviewer for the insightful comment. As requested, we have now included in the manuscript robust scientific evidence regarding the negative effects of the half-time break on the specific variables investigated in our study. Specifically, we added references demonstrating decreases in sprint and jump performance following passive rest (Greig & Siegler, 2009; Bang & Park, 2022; Christaras et al., 2023), as well as literature linking reductions in muscle temperature to lower subjective readiness scores (Mohr et al., 2004; Russell et al., 2015), and findings from the ARMS questionnaire assessing subjective muscle readiness. These references have been integrated into the relevant sections of the article to reinforce the theoretical foundation of our work and to support the rationale for implementing re-warm-up protocols.
Comment 2 – “Three-minute re-warm-up activities are effective in high-intensity exercises, but there is no evidence of their effectiveness in prolonged activities such as soccer.”
Response: We appreciate the reviewer’s suggestion. The rationale for selecting the two rewarm-up protocols has been clarified and expanded in the revised Introduction. The SPBP protocol was adapted from the FIFA 11+ program and is grounded in neuromuscular and proprioceptive activation (Bizzini et al., 2013; Barengo et al., 2014; Tillin & Bishop, 2009), aiming to elicit post-activation potentiation and improve joint stability (Ekstrand et al., 2011). In contrast, the SSP was based on the use of small-sided games to replicate contextual match demands (Hill-Haas et al., 2011; Zois et al., 2013), stimulating physiological, cognitive, and technical engagement (Abade et al., 2017). These theoretical foundations and expected effects have now been explicitly included in the Introduction. We also agree with the reviewer’s point regarding the lack of a passive control condition. As noted in the original version of the manuscript, we acknowledged this limitation and justified our choice by referring to prior studies that consistently demonstrate the negative effects of passive rest on second-half performance (Mohr et al., 2004; Weston et al., 2011; Hammami et al., 2018). This point remains clearly stated in the revised manuscript.
Methods:
Comment 1 – “In addition to previous statement, before making comparisons, it's important to evaluate the effectiveness of the proposed programs. This highlights the necessity for a control group (a group with a passive break). In the proposed design, we are only observing the differences between the programs, but we cannot determine whether these programs are effective at all. There is no previous evidence either.”
Response: We appreciate the pertinent observation, this study was exploratory in nature, aiming primarily to assess the feasibility and effects of applying these two distinct rewarm-up protocols during halftime. Interestingly, despite their differing characteristics, both protocols produced similar results in our sample. Due to the convenience sampling and the need to minimize interference with the regular training and competitive processes of this high-level team, it was not feasible to divide participants into three groups, and therefore, a passive control group was not included. We acknowledge this limitation, which was already clearly stated in the original version of the manuscript. However, the objective of this study was to compare two re-warm-up protocols that are already supported by previous literature regarding their effectiveness in mitigating performance decline after halftime, although they operate through distinct mechanisms. We opted not to include a passive control group because several studies in the literature have consistently demonstrated the negative effects of passive rest on physical and technical-tactical performance at the start of the second half (Mohr et al., 2005; Weston et al., 2011; Hammami et al., 2018), which allowed us to focus our investigation on the direct comparison between two active interventions. Nonetheless, we recognize the value of future studies incorporating a control condition to more precisely quantify the magnitude of each protocol’s effects relative to the absence of intervention.
Comment 2 – “In my opinion, the authors should first evaluate the effectiveness of each program individually and validate the ARMS questionnaire that is being used.”
Response: We appreciate your insightful comment. However, we would like to clarify that the primary aim of this study was not to validate the ARMS questionnaire nor to assess the isolated effectiveness of each re-warm-up protocol. Our main objective was to compare two distinct and time-efficient re-warm-up strategies (SPBP and SSP), both of which are supported in the literature for their potential to mitigate post-halftime performance decline via different physiological mechanisms. The ARMS questionnaire served only as a secondary, exploratory outcome with limited emphasis in our discussion, aimed at providing an additional subjective perspective on players’ perceived readiness.
The SPBP protocol is grounded in neuromuscular and proprioceptive activation (Bizzini et al., 2013; Tillin & Bishop, 2009; Ekstrand et al., 2011), whereas the SSP protocol aims to replicate the game-specific demands and promote technical-cognitive engagement (Zois et al., 2013; Hill-Haas et al., 2011; Abade et al., 2017). Therefore, this study was designed to comparatively assess which approach might be more effective under the practical constraint of a three-minute halftime window within a competitive setting.
Regarding the ARMS questionnaire, we acknowledge that it is still undergoing validation in the scientific literature. Nonetheless, its use in this study served a complementary purpose alongside objective performance measures (i.e., sprint and vertical jump), aiming to capture players’ subjective perception of readiness in a quick and practical manner. We agree that future investigations should further address the validation of the ARMS scale across diverse athletic populations.
Comment 3 - Is there a randomization process for the order of testing procedures?
Response: The re-warm-up protocols (SPBP and SSP) were randomly assigned using simple randomization. However, the testing procedures followed a fixed order for all participants to ensure standardization and minimize variability.
Comment 4 – “lines 218-220: The presented description is for the Squat jump, not for the Countermovement?”
Response: We apologize for the oversight. To perform the CMJ, participants were asked to stand upright, place their hands on their hips, and keep them there throughout the test. When ready, the athlete squatted down until the knees were bent at a 90-degree angle, then immediately jumped vertically as high as possible.
Comment 5 – “Why didn’t the authors perform the ARMS questionnaire before the match for comparison, similar to the Sprint and Jump tests?”
Response: We chose not to administer the ARMS questionnaire before the match because doing so could influence and diminish the effects captured by the questionnaire after the re-warm-up and implementation of the warm-up protocols. Furthermore, as indicated by several studies, readiness questionnaires are more appropriately applied after the execution of specific protocols, as this better reflects the athlete’s actual state of preparation. We also considered the logistical challenges of ensuring all athletes would be available to complete the questionnaire before the game, especially given the preparations and dynamics involved in simulating the match. Therefore, administering the ARMS questionnaire after the warm-up protocols allows for a more accurate assessment of task readiness, reflecting the athletes’ true preparation status and considering the specific game dynamics.
Comment 6 – “Comparing post-warm-up and post-re-warm-up differences is meaningless due to fatigue factors; the difference is obvious.”
Response: We respectfully disagree. While the presence of fatigue is expected, differences in the magnitude of performance decline between protocols provide relevant information on their relative effectiveness. If one protocol mitigates the drop more than the other, that comparison is both meaningful and informative.
Results:
Comment 1 – “A graphical presentation of the results would be helpful.”
Response: We appreciate the suggestion to include graphical representations of the results. While we initially opted for tables, which we considered clear and sufficient for effectively conveying the findings, we agree that figures can enhance data interpretation. Therefore, we have included graphical representations alongside the tables in the revised version of the manuscript.
Discussion:
Comment 1 – “In the Discussion section, the authors primarily reiterated data from the Results and Introduction without providing a deeper explanation of the findings.”
Response: The discussion interprets the results in light of existing literature and contextualizes them within previous findings. We took this opportunity to improve the manuscript by expanding the comparative analysis between protocols and integrating additional references to support a more robust discussion.
Comment 2 – “The limitation of the study regarding its sample size is valid; however, the absence of a control group is a significant oversight by the author. A better approach would be to have three groups of seven players each, as this would allow us to assess the effectiveness of the program more effectively. Regardless, the sample size remains small for both two and three groups, which is why this research is consider as a pilot study. Additionally, a pilot study should serve as a preliminary step for larger research; however, the results of this study do not provide a clear direction for the next steps.|”
Response: Dividing the participants into three groups of seven would have dramatically reduced the statistical power of our study, making it unlikely to detect any meaningful effects unless they were exceptionally large. Our design focused on comparing two active protocols, a choice that was informed by previous research (e.g., Edholm et al., 2014; Yanaoka et al., 2020) and was optimal given the practical and logistical constraints. We agree that this is a pilot study, and indeed, its main value lies in showing that two quite distinct re-warm-up protocols produced similar outcomes in sprint, jump, and readiness variables. This suggests that the specific exercises chosen during a short halftime re-warm-up may be less critical than previously assumed, potentially granting coaches greater flexibility when designing practical strategies. Therefore, our results serve as a basis for the formulation of a hybrid protocol that can be evaluated in future studies, possibly with larger samples and control groups, to determine its efficacy and practical benefits more comprehensively. This interpretation has now been made more explicit and clearly stated in the revised version of the Discussion section
Final conclusion:
Comment 1 – “Although the authors define their study as a pilot study, there should be strong findings that justify publication.”
Response: To avoid publication bias, all results—regardless of their direction or magnitude—should be eligible for publication. Restricting publication to only “strong” or statistically significant findings contributes to a well-documented bias in the scientific literature, an issue that has been widely criticized across various fields, including sports science (Munafò et al., 2017; Ioannidis, 2005; Nosek et al., 2015; Halperin et al., 2018).
Pilot and exploratory studies are, by design, not intended to produce confirmatory or “strong” results. Instead, their purpose is to generate preliminary evidence, assess feasibility, and identify key parameters for future research. Such studies provide valuable insights and help guide the development of larger, adequately powered confirmatory trials where robust conclusions can be drawn. The primary goal is to explore and refine research questions and methodologies rather than to confirm hypotheses outright (Thabane et al., 2010; Rounis et al., 2021; Leon et al., 2011; Arain et al., 2010).
Our study aligns with this exploratory framework by offering ecologically valid, comparative data on two distinct re-warm-up protocols applied under realistic soccer match conditions. While strong effects were neither expected nor observed, these preliminary findings contribute important insights to an underexplored area and establish a foundation for future, more definitive research involving larger sample sizes and control groups. Although characterized as a pilot study due to its sample size and exploratory nature, it provides valuable preliminary insights into the comparative effectiveness of two practical re-warm-up protocols within the constrained halftime period typical of soccer matches. The findings contribute meaningful data on key performance metrics (sprint and countermovement jump) and perceived readiness—areas that remain underexamined in this applied context. Furthermore, the ecological validity of the study, achieved by simulating real match conditions, enhances the practical relevance of the results. These preliminary findings offer an essential basis for designing more robust future investigations and support the optimization of re-warm-up protocols tailored to the specific demands of soccer players.
Similar pilot studies focusing on re-warm-up interventions (e.g., Edholm et al., 2014; Yanaoka et al., 2020) have served as crucial initial steps in establishing effective strategies, justifying their significance despite limited sample sizes. Accordingly, despite its pilot status, our study offers meaningful contributions that warrant publication and inclusion in the literature. As argued by Impellizzeri et al. (2019) in their discussion on the importance of studies with null or inconclusive results, the publication of these works is fundamental to avoid publication bias and provide a more complete view of the body of scientific knowledge, especially in applied areas such as sports sciences.
Comment 2 – “The main finding of this study is that there is no difference among the evaluated programs based on any of the assessed variables.”
Response: We agree with the reviewer’s observation: yes, it is true that the two active re-warm-up protocols (SSP and SPBP) resulted in similar outcomes in sprint, jump performance, and perceived readiness. This means that, despite appearing quite distinct in design and content, both re-warm-up strategies produced comparable effects on the analyzed variables. This is a relevant finding, as it suggests that different approaches—whether more neuromuscular or more sport-specific—may offer similar benefits in the context of a short halftime re-warm-up. This interpretation aligns with previous work showing that various re-warm-up strategies can yield equivalent outcomes, depending on timing and context (Edholm et al., 2014; Yanaoka et al., 2020). Although the absence of a passive control group limits conclusions regarding absolute effectiveness in preventing performance decline, our goal was to compare the relative impact of these two practical and time-efficient protocols. These findings provide practical guidance for coaches and trainers, indicating flexibility in selecting re-warm-up methods based on context and available resources, emphasizing the need to adjust protocols to balance benefits and potential adverse effects on athletic performance. Future research should consider regulatory factors and explore implementation feasibility in official match settings.
Comment 3 – “Given the earlier statement that there is no evidence supporting the effectiveness of the two proposed programs, we must ask how the publication of this manuscript can be justified.”
Response: To avoid publication bias, all results—regardless of their direction or magnitude—should be eligible for publication. Restricting publication to only “strong” or statistically significant findings contributes to a well-documented bias in the scientific literature, an issue that has been widely criticized across various fields, including sports science (Munafò et al., 2017; Ioannidis, 2005; Nosek et al., 2015; Halperin et al., 2018).
This study addresses a practical and highly relevant issue in the context of soccer by exploring re-warm-up strategies within the limited halftime period—an area with direct applicability for coaches and sports performance professionals. The lack of difference between protocols suggests flexibility for practitioners in choosing or combining approaches, which is valuable practical information. This flexibility is particularly important considering the time constraints during halftime, as highlighted by Towlson et al. (2013), who reported that only 58% of Premier League and Championship coaches implement re-warm-up programs, with lack of time being the main reason for not using these protocols. Our study aligns with this exploratory framework by offering ecologically valid, comparative data on two distinct re-warm-up protocols applied under realistic soccer match conditions. While strong effects were neither expected nor observed, these preliminary findings contribute important insights to an underexplored area and establish a foundation for future, more definitive research involving larger sample sizes and control groups.
Although characterized as a pilot study due to its sample size and exploratory nature, it provides valuable preliminary insights into the comparative effectiveness of two practical re-warm-up protocols within the constrained halftime period typical of soccer matches. The findings contribute meaningful data on key performance metrics (sprint and countermovement jump) and perceived readiness—areas that remain underexamined in this applied context. Furthermore, the ecological validity of the study, achieved by simulating real match conditions, enhances the practical relevance of the results. These preliminary findings offer an essential basis for designing more robust future investigations and support the optimization of re-warm-up protocols tailored to the specific demands of soccer players. This is particularly important given that much of the current literature relies on laboratory-based protocols that do not always reflect the constraints and demands of actual sports settings. As highlighted in the recent systematic review with meta-analysis by González-Fernández et al. (2023), there is a critical need for studies with greater ecological validity that assess performance in conditions that approximate real game situations. The authors emphasize that "more high-quality studies, with homogeneous study designs, may help to clarify the potential benefits of re-warm-up for linear sprint time," underlining the importance of preliminary studies such as ours.
Similar pilot studies focusing on re-warm-up interventions (e.g., Edholm et al., 2014; Yanaoka et al., 2020) have served as crucial initial steps in establishing effective strategies, justifying their significance despite limited sample sizes. Edholm et al. (2014), investigated the acute effects of a halftime re-warm-up on performance and movement patterns in soccer, Yanaoka et al. (2018), who examined the effects of a just 1-minute high-intensity re-warm-up on sprint performance, and Christaras et al. (2023), who evaluated a short re-warm-up program at halftime in elite youth players—have also been published despite small sample sizes and inconclusive findings, demonstrating the value of preliminary work in this field. Particularly and highlight that "the short re-warm-up program can limit the decrement in performance in power tests such as sprints and jumps," even without significant differences in all measured variables. This observation aligns with our results and reinforces the importance of publishing studies that provide practical information for professionals, even when the results do not show clear superiority of one protocol over another. Sanchez et al. (2023) recently published a study on the use of elastic bands during half-squats as a re-warm-up strategy, even with methodological limitations similar to those in our work.
As a pilot study, these findings offer a solid foundation for future larger-scale and controlled research. This includes the potential development of hybrid re-warm-up protocols combining the benefits of different neuromuscular and sport-specific exercises, which may better address the complex demands of soccer performance. Publishing this work encourages ongoing investigation and critical discussion regarding best practices in re-warm-up strategies, ultimately advancing sports science.
Therefore, despite the lack of conclusive evidence favoring one protocol over the other, this study holds both scientific and practical merit that justifies its dissemination.
As argued by Impellizzeri et al. (2019) in their discussion on the importance of studies with null or inconclusive results, the publication of these works is fundamental to avoid publication bias and provide a more complete view of the body of scientific knowledge, especially in applied areas such as sports sciences.
Reviewer 3 Report
Comments and Suggestions for Authors
I thank the authors and the editorial office for giving me the opportunity to review this study. The present manuscript provides a simple – yet relevant – investigation into the effect of two re-warm-up protocols performed at halftime in female soccer. The study is generally well-written and easy to follow. However, several aspects need to be addressed before the manuscript is suitable for publication.
My comments can be found below.
INTRODUCTION
L 37-40: As described in the review by McGowan et al. (https://pubmed.ncbi.nlm.nih.gov/26400696/), I suggest that you briefly expand on the mechanisms responsible for warm-up induced improvements in performance. Specifically, increased muscle temperature leads to increased muscle metabolism, (which results in quicker ATP turnover) and higher muscle fibre conduction velocity. All together, these changes lead to increased power output. Furthermore, faster VO2 kinetics are also among the mechanisms involved in warm-up induced performance improvements.
L 40-68: The overall presentation and logical flow of this section is a bit messy. At the beginning of this part, re-warm-up strategies are mentioned “out of the blue”, without a clear introduction. I think that the best option would be to start with the section which is currently at Lines 46-54 (where you provide a nice overall characterization of warm-ups in football), and then moving to the specifically discussing re-warm-ups. I do not believe that you need to perform major changes here (the content is ok), but rather just re-arrange the paragraphs within this section (to provide a better logical flow).
L 61: I suggest removing “a”: “result in decreased body temperature”
L 64-67: I am curious to know how the first 15 min of the second half compare with the last part of the match (where I would expect fatigue to considerably decrease intensity). Can you confirm that the start of the second half is even less intense than match-end? If not, this sentence may need to be slightly re-structured.
L 69: It appears that the authors’ name is missing here. After looking at the reference list, I believe this should be: “The study by Sanchez-Sanchez et al. [15]”
L 77: By “agility”, do you mean actual agility drills (movements which include a decision-making component), or simple “change-of-direction” drills (which do not involve such component)? Please refer to the categorization provided by Paul et al. (https://pubmed.ncbi.nlm.nih.gov/26670456/), and ensure you are use the correct terminology. The same comment is valid at Line 127.
L 90-91: You mention “These authors” here. Which authors are you referring to, exactly?
L 104-107: Please specify the duration of warm-up used in this study. Furthermore, it feels like the description of this study would fit better in the previous paragraph (Lines 93-99), where you are presenting short-duration protocols.
METHODS
L 122-125: Although you found no relevant difference between groups, I am a bit surprised to see that you did not utilize a crossover design. That seemed very feasible in this context, and it would have substantially contributed to increasing the quality of your study. Please add this aspect to the study limitations at the end of the discussion.
L 125-130: This part should go somewhere in section 2.3 (“Interventions”), or 2.3.1 (where you specifically describe SPBP). To avoid redundancies, it is sufficient to present the overall idea in section 2.1 (“Study design”), by naming the interventions (as you already did at Lines 130-132), and also including a brief overview of all procedures utilized in the study and their sequence (and, if needed, other general considerations which are not strictly tied to any specific procedure).
L 145-146: Please clarify the meaning of this sentence.
Figure 2: Please correct the spelling of “Radomized” to “Randomized”. I also suggest changing “SSP group” to just “SSP”.
L 161-162: Can you please share the structure of the “usual” team warm-up (i.e., tasks, duration of each task, total duration, etc)? You can include a table/figure within as supplementary material.
L 170: I suggest replacing “they” with “participants”. I also think the format “(1+10 vs 1+10)” is a bit confusing. I suggest removing the brackets, and maybe just saying “11 vs 11” or something like that. Finally, you do not explicitly mention whether goalkeepers were included (although I suspect that is what “1+10” means).
L 172-173: It is clear that you only used the last 3 minutes for the re-warm-up. What happened during the previous 12 minutes (i.e., from the end of the first half, until the re-warm-up)? This is an important aspect of your study, as any activity performed prior to the re-warm-up start would negatively affect the quality of your data. What were the participants instructed to do?
L 182: There is a typing mistake in this sentence: “enhance performance in soccer players through soccer players through specific strength”. Please correct.
L 203-229: I suggest that you merge these two paragraphs (20m and CMJ) into just one paragraph, and name it “Performance assessment”.
L 204-206: It is currently unclear what type of “speed” you analyzed here. Was it the average speed during the 20m sprint? Or was it the top speed? Furthermore, please describe the procedure that you used to cut data on S-PRO (in order to isolate each 20m sprint for each player, and make sure that the speed measured was actually related to the specific sprint performed by each participant).
L 226-229: Using a single sprinting and jumping trial to assess performance presents a number of issues. Crucially, using a single trial does not ensure that true maximal performance is detected, as athletes may underperform during the single attempt (e.g., due to insufficient motivation, technical mistakes or inconsistent execution). That can considerably influence the accuracy and reliability of CMJ and sprint measurements. To mitigate this issue, previous research has largely utilized multiple attempts for jump and sprint testing (at least 2), either considering the average value or best result (Moir et al., 2004: https://pubmed.ncbi.nlm.nih.gov/15142028/; Kamandulis et al., 2013: https://pubmed.ncbi.nlm.nih.gov/24665800/; Attia et al., 2017: https://pubmed.ncbi.nlm.nih.gov/28416900/; Mercer et al., 2023: https://pubmed.ncbi.nlm.nih.gov/36696261/). Therefore, this limitation needs to be added in section 4.1, and the discussion needs to clearly state that the results from this study should be interpreted with caution.
L 231-232: Looking at Figure 1, it seems like ARMS was only administered to players before the start of the 2nd half. If so, why wasn’t it also administered right before the match (i.e., after CMJ and sprint tests were performed)? That way, you could have compared perceived readiness after the “regular” warm-up (i.e., pre-match) with readiness after the re-warm-up (at halftime).
RESULTS
L 260: Please rephrase as “There was no significant difference in CMJ […]”
Table 1: In the methods section (Line 205), it is stated that you measured speed (rather than time) associated with each 20m sprint. However, here you report sprint results in seconds (rather than km/h or m/s, as one may expect). This needs to be clarified, and a consistent characterization of sprint-related metrics needs to be presented in the methods and results sections.
Table 2: Are the terms “Specific” and “Power” synonyms of “SSP” and “SPBP”, respectively? If so, I suggest using the abbreviated version here too (to maintain consistent terminology and avoid confusion).
DISCUSSION
General comments
Overall, the discussion does not include considerations regarding the mechanisms underlying the two different protocols (i.e., SSP and SPBP), and what effects such mechanisms could have in terms of performance enhancement. For example, the mechanisms elicited by strength-related exercises and plyometrics could be mainly related to PAP-induced benefits (or similar), while SSGs may primarily result in increased muscle temperature. The ones I offered above are just examples, but you need to conduct a more thorough analysis of which factors are responsible for the effectiveness of each method. To do so, you may look at the review by McGowan et al. (https://pubmed.ncbi.nlm.nih.gov/26400696/). After this has been done, you can move on and discuss your specific findings (and how they compare to previous research, as you already did).
Furthermore, when comparing your findings with previous research, it is important that you describe the characteristics of interventions employed in other studies. For example, different warm-up durations, intensities and exercise modalities (e.g., cycling, running, weight lifting, etc.) are factors that could explain differences between your findings and those in previous research.
Specific comments
L 300-302: You make a very good point here, regarding the importance of ecological validity. However, I suggest that you slightly expand, and add supporting literature here. Here is my suggestion: “Although the intervention was conducted in a simulated match environment, the protocols were designed to replicate the constraints and demands of an official soccer match, enhancing ecological validity and supporting the translation of findings to real-world scenarios (*add reference*).” As a supporting reference, you can use the following study by Pernigoni et al. (2024): https://pubmed.ncbi.nlm.nih.gov/39179229/
L 302-304: Based on your findings, you cannot make a general statement such as “our findings demonstrate that it is possible to implement effective re-warm-up protocols”. Considering that one of your two measures (i.e., 20m sprint times) displayed significantly worse values after the re-warm-up (compared to pre-match), this indicates that the type of re-warm-up used here was not entirely able to restore pre-match performance. Admittedly (as you state in section 4.1), there may be other factors that influenced your results, such as the possible presence of fatigue after the first half (see Silva et al., 2018: https://pubmed.ncbi.nlm.nih.gov/29098658/), which could result in slower sprint times. Nevertheless, you also need to consider the possibility that a 3-min re-warm-up may not have been sufficient to adequately re-prepare the players for the second half. Accordingly, a previous systematic review (Silva et al., 2018: https://pubmed.ncbi.nlm.nih.gov/29968230/) indicates that 5-8 minutes may be needed for an adequate re-warm-up. Therefore, while a 3-minute re-warm-up may be more feasible in real-world settings (which I completely understand), this does not change the fact that it may not have been entirely effective. Accordingly, you need to correct the overall message of your discussion (both here, and also at Lines 336-343) and conclusions.
L 313-316: This sentence needs to be restructured (it is currently a bit messy in terms of lexical clarity).
L 317: I suggest rephrasing as follows: “[…] in both protocols (although differences did not reach statistical significance), with a slightly larger [...].”
L 319-320: After reporting conflicting findings from previous research, it seems like you just moved on without discussing why such differences may exist. This part needs to be expanded, by providing potential reasons as to why these differences are present. Could it be due to different re-warm-up protocols being used? I suspect that a different duration, type of exercise and intensity could – potentially – explain such differences (if it is confirmed that such differences exist among previous research). Or maybe there could be some other reasons?
L 321-324: At Line 322, you should specify that this is compared to pre-match measurements. For example: “[…] following the re-warm-up protocols, compared to post-warm-up (i.e., pre-match)”. Similarly, when discussing findings from Yanaoka et al., you also need to specify that (i.e., “improvements after a re-warm-up”, but compared to what time point?). Furthermore, it is unclear how the re-warm-up by Yanaoka et al. was structured. What does “either one minute […] or three minutes” indicate? Did they use 2 different protocols, or was it one protocol (since you state “re-warm-up program” at Line 323)?
L 324-328: This sentence does not logically follow the previous one. How can it be that Yanaoka et al. reported improvements in sprint performance following re-warm-up (as you state in the previous sentence), but simultaneously detected a decline in sprint performance (as you state here)? I think that you might have mixed your own findings with those of Yanaoka et al. (considering that mean and SD values reported for SSP and SPBP are the same as Table 1). If so, this sentence needs to be corrected, and it should be moved to Line 322. I also noticed that values reported here differed from what was reported in Table 1 (it seems there was some confusion between post-warm-up and post-re-warm-up values). Therefore, the sentence may be rephrased as follows (starting at Line 321): “Our results indicated a significant decline in sprint performance following the re-warm-up protocols. Specifically, we noted worse sprint performance after the re-warm-up (SSP: 3.86±0.15; SPBP: 3.88±0.26), compared with post-warm-up measures assessed at pre-match (SSP: 3.51±0.15; SPBP: 3.88±0.26). Conversely, Yanaoka et al. […]”
L 328-335: Again, you need to indicate which time point was compared to “after halftime” in this study. Did they compare pre-match with post-halftime? If so, were pre-match measurements taken after warm-up? Please specify.
L 347-348: It seems like this sentence is incomplete.
L 363-364: Please add supporting references for this statement: “[…] could be largely attributed to accumulated fatigue during the match (*add references*), without providing […]. Silva et al. (2018): https://pubmed.ncbi.nlm.nih.gov/29098658/; Pernigoni et al. (2024): https://pubmed.ncbi.nlm.nih.gov/39374409/
ABSTRACT AND CONCLUSIONS
These sections will be reconsidered once edits have been applied to the rest of the manuscript.
Author Response
Dear Reviewer,
We are grateful for the insightful comments and suggestions provided, which have been invaluable in improving our work. We have carefully addressed all the feedback and have highlighted the revisions in the manuscript as requested.
INTRODUCTION
Comment 1 – “L 37-40: As described in the review by McGowan et al. (https://pubmed.ncbi.nlm.nih.gov/26400696/), I suggest that you briefly expand on the mechanisms responsible for warm-up induced improvements in performance. Specifically, increased muscle temperature leads to increased muscle metabolism, (which results in quicker ATP turnover) and higher muscle fibre conduction velocity. All together, these changes lead to increased power output. Furthermore, faster VO2 kinetics are also among the mechanisms involved in warm-up induced performance improvements.”
Response: As suggested, we have now included this explanation in the revised manuscript to clarify how warm-up contributes to performance improvements.
Comment 2 – “L 40-68: The overall presentation and logical flow of this section is a bit messy. At the beginning of this part, re-warm-up strategies are mentioned “out of the blue”, without a clear introduction. I think that the best option would be to start with the section which is currently at Lines 46-54 (where you provide a nice overall characterization of warm-ups in football), and then moving to the specifically discussing re-warm-ups. I do not believe that you need to perform major changes here (the content is ok), but rather just re-arrange the paragraphs within this section (to provide a better logical flow).”
Response: Thank you for this helpful comment. We fully agree with your suggestion regarding the reorganization of the section between Lines 40–68. Following your advice, we have moved the general characterization of warm-ups in football (originally at Lines 46–54) to the beginning of this section. This adjustment indeed provides a clearer context and improves the logical flow of the text, allowing the subsequent discussion on re-warm-up strategies to be more coherent and easier to follow.
Comment 3 – “L 61: I suggest removing “a”: “result in decreased body temperature”
Response: Removed accordingly.
Comment 4 – “L 64-67: I am curious to know how the first 15 min of the second half compare with the last part of the match (where I would expect fatigue to considerably decrease intensity). Can you confirm that the start of the second half is even less intense than match-end? If not, this sentence may need to be slightly re-structured.”
Response: Thank you for your insightful comment regarding the intensity of the first 15 minutes of the second half. We agree this is an important point and have revised the sentence accordingly. Studies have shown that this period is often less intense than expected, likely due to reductions in muscle temperature and neuromuscular readiness following the passive half-time break (Mohr et al., 2003; Russell et al., 2016; Lovell et al., 2013). We have clarified this point in the manuscript by incorporating this rationale and citing the relevant studies. We believe this adjustment enhances the interpretation of performance fluctuations across different match periods.
Comment 5 – “L 69: It appears that the authors’ name is missing here. After looking at the reference list, I believe this should be: “The study by Sanchez-Sanchez et al. [15]”
Response: Corrected as requested.
Comment 6 – “L 77: By “agility”, do you mean actual agility drills (movements which include a decision-making component), or simple “change-of-direction” drills (which do not involve such component)? Please refer to the categorization provided by Paul et al. (https://pubmed.ncbi.nlm.nih.gov/26670456/), and ensure you are use the correct terminology. The same comment is valid at Line 127.”
Response: Thank you for this important observation. You are right in highlighting the need to distinguish between “agility” and “change-of-direction” (COD) drills, based on the categorization proposed by Paul et al. (2016). After revising the terminology used in our manuscript, we acknowledge that the drills described in the referenced study (related to the re-warm-up intervention) do not include a perceptual or decision-making component. Therefore, the correct term to use is “change-of-direction” drills rather than “agility”.
Comment 7 – “L 90-91: You mention “These authors” here. Which authors are you referring to, exactly?”
Response: The intended reference was Towlson et al. (2013). We have revised the sentence to explicitly mention the authors: "Towlson et al. (2013) reported that time constraints and coaches' reluctance limit the opportunity for re-warm-up in official soccer matches, posing significant barriers to the implementation of pre-match re-warm-up procedures."
Comment 8- “L 104-107: Please specify the duration of warm-up used in this study. Furthermore, it feels like the description of this study would fit better in the previous paragraph (Lines 93-99), where you are presenting short-duration protocols.”
Response: The total warm-up duration was approximately 15 to 20 minutes. We have now included this information in the manuscript. Furthermore, we have moved this description to the previous paragraph (Lines 93–99) to maintain logical consistency with the discussion of short-duration protocols.
METHODS:
Comment 1- “L 122-125: Although you found no relevant difference between groups, I am a bit surprised to see that you did not utilize a crossover design. That seemed very feasible in this context, and it would have substantially contributed to increasing the quality of your study. Please add this aspect to the study limitations at the end of the discussion.”
Response: We agree with the reviewer, but practical constraints inhibited the implementation of crossover design. Specifically, it would require greater interference with the team’s natural training process. Therefore, we decided not to use a crossover design, despite the limitations of doing so. We now acknowledge this in the manuscript.
Comment 2- “L 125-130: This part should go somewhere in section 2.3 (“Interventions”), or 2.3.1 (where you specifically describe SPBP). To avoid redundancies, it is sufficient to present the overall idea in section 2.1 (“Study design”), by naming the interventions (as you already did at Lines 130-132), and also including a brief overview of all procedures utilized in the study and their sequence (and, if needed, other general considerations which are not strictly tied to any specific procedure).”
Response: Thank you for your insightful suggestion. We agree with your observation that the current placement of this content creates redundancy with the overall structure of the manuscript. Presenting the general idea and overview of procedures in Section 2.1 (“Study Design”), while keeping the specific details within the relevant subsections (e.g., 2.3 and 2.3.1), will indeed enhance the clarity and logical flow of the methods section. We have revised the manuscript accordingly to avoid repetition and improve organization. However, we had to add additional information at the request of other reviewers.
Comment 3- “L 145-146: Please clarify the meaning of this sentence.”
Response: Rephrased: “The study included female soccer players with no clinical limitations for playing the match”, which we believe better clarifies the inclusion criteria.
Comment 4- “Figure 2: Please correct the spelling of “Radomized” to “Randomized”. I also suggest changing “SSP group” to just “SSP”
Response: Corrected accordingly.
Comment 5- “L 161-162: Can you please share the structure of the “usual” team warm-up (i.e., tasks, duration of each task, total duration, etc)? You can include a table/figure within as supplementary material.”
Response: Thank you for your suggestion. Unfortunately, we are unable to provide the detailed structure of the “usual” team warm-up, including tasks and their durations, as this information is confidential and we do not have authorization from the coaching staff to disclose it. We appreciate your understanding regarding this matter. Moreover, as all players performed the same warm-up, this was not a differentiating factor between groups.
Comment 6- “L 170: I suggest replacing “they” with “participants”. I also think the format “(1+10 vs 1+10)” is a bit confusing. I suggest removing the brackets, and maybe just saying “11 vs 11” or something like that. Finally, you do not explicitly mention whether goalkeepers were included (although I suspect that is what “1+10” means).”
Response: Thank you for your valuable suggestion. The “1+10” format was somewhat confusing. Our intention was to indicate that goalkeepers were included in the study. We will revise the text to replace “they” with “participants,” remove the brackets, and clearly state “11 vs 11” to improve clarity.
Comment 7- “L 172-173: It is clear that you only used the last 3 minutes for the re-warm-up. What happened during the previous 12 minutes (i.e., from the end of the first half, until the re-warm-up)? This is an important aspect of your study, as any activity performed prior to the re-warm-up start would negatively affect the quality of your data. What were the participants instructed to do?”
Response: We appreciate your insightful question regarding the activities performed during the 12 minutes prior to the re-warm-up. During this period, the athletes remained in passive rest in the locker room. They were instructed not to engage in any physical activity that could compromise the study’s outcomes. Participants were allowed to perform normal halftime activities such as using the restroom, listening to the coach’s instructions, and hydrating, but no additional physical exertion was permitted. Any deviation from these instructions would have been reported and considered in the analysis. This has now been reported in the manuscript.
Comment 8- “L 182: There is a typing mistake in this sentence: “enhance performance in soccer players through soccer players through specific strength”. Please correct.”
Response: Corrected accordingly.
Comment 9- L 203-229: I suggest that you merge these two paragraphs (20m and CMJ) into just one paragraph, and name it “Performance assessment”.
Response: We have implemented this change in the revised manuscript accordingly.
Comment 10- L 204-206: It is currently unclear what type of “speed” you analyzed here. Was it the average speed during the 20m sprint? Or was it the top speed? Furthermore, please describe the procedure that you used to cut data on S-PRO (in order to isolate each 20m sprint for each player, and make sure that the speed measured was actually related to the specific sprint performed by each participant).
Response: Thank you for your observation. We clarified that the analysis was based on the average speed recorded during the 20-meter sprint test. The data were collected using WIMU PRO™ GPS devices (RealTrack Systems, Almería, Spain), which were assigned individually to each player. To ensure accuracy, the beginning and end of each 20-meter sprint were identified using the GPS tracking data, and each sprint was individually tracked and isolated through the S-PRO™ analysis software. This allowed us to determine the average speed over the exact 20-meter distance for each athlete.
The same procedure was applied in both evaluation moments (i.e., post-warm-up and post-re-warm-up), enabling a direct and reliable comparison between the two time points for each player. We revised the manuscript to include these clarifications and ensure that the type of speed analyzed and the data extraction method are explicitly described.
Comment 11- L 226-229: Using a single sprinting and jumping trial to assess performance presents a number of issues. Crucially, using a single trial does not ensure that true maximal performance is detected, as athletes may underperform during the single attempt (e.g., due to insufficient motivation, technical mistakes or inconsistent execution). That can considerably influence the accuracy and reliability of CMJ and sprint measurements. To mitigate this issue, previous research has largely utilized multiple attempts for jump and sprint testing (at least 2), either considering the average value or best result (Moir et al., 2004: https://pubmed.ncbi.nlm.nih.gov/15142028/; Kamandulis et al., 2013: https://pubmed.ncbi.nlm.nih.gov/24665800/; Attia et al., 2017: https://pubmed.ncbi.nlm.nih.gov/28416900/; Mercer et al., 2023: https://pubmed.ncbi.nlm.nih.gov/36696261/). Therefore, this limitation needs to be added in section 4.1, and the discussion needs to clearly state that the results from this study should be interpreted with caution.
Response: We appreciate the reviewer’s valuable comments regarding the use of a single trial for the sprint and jump tests. While we acknowledge that multiple trials are often employed to ensure maximal performance, we respectfully argue that, within the specific context of our study, the use of a single attempt is methodologically justified and does not compromise the validity of our results.
Firstly, all athletes participated in a prior familiarization session, during which they were introduced to the re-warm-up protocols and the performance assessment procedures. This step is well-supported in the literature as a means to reduce inter-individual variability and ensure technical consistency in subsequent evaluations (Hopkins et al., 2001). On the test day, players were explicitly informed that only one attempt would be allowed for each test. This advance notice heightened their mental preparation and focus, helping to ensure that they gave their best effort during that single trial and reducing the risk of underperformance due to lack of motivation or concentration.
It is also important to emphasize that our participants were professional athletes with extensive experience in physical performance testing. Prior studies (e.g., Moir et al., 2004) have demonstrated that elite athletes exhibit greater consistency and lower variability across trials compared to non-trained individuals, thereby reducing the necessity of multiple attempts to obtain reliable data.
A critical component of our experimental design was the need to assess performance within the specific time window during which the physiological effects of the re-warm-up are most pronounced. Implementing multiple trials would have required rest intervals that could potentially diminish or obscure the acute effects of the re-warm-up, as noted by Yanaoka et al. (2018). Thus, to preserve the ecological validity and temporal sensitivity of our design, performance was assessed immediately after the re-warm-up using a single trial.
Moreover, our approach is supported by previous literature in the field. For instance, Christaras et al. (2023) used a single trial to assess 10-m and 30-m sprints, agility, countermovement jumps (CMJ), and squat jumps (SJ) in elite youth soccer players following a re-warm-up protocol. Similarly, Yanaoka et al. (2018) employed single intermittent sprints to assess performance post-intervention, and Mohr et al. (2004) also relied on single sprint assessments in professional football players. These examples confirm that the use of single attempts in re-warm-up studies is a widely accepted and methodologically sound approach.
Finally, our experimental design was intentionally structured to maximize ecological validity and real-world applicability. The use of a single trial mirrors actual match conditions, where players do not have the luxury of multiple attempts to execute explosive actions. This choice strengthens the relevance of our findings for practical application in competitive sport settings. Furthermore, rigorous control of testing conditions, prior familiarization, and standardized procedures ensured high data quality and minimized measurement error.
In light of these considerations, we maintain that the use of a single trial for sprint and jump tests was both deliberate and appropriate for the objectives of our study. Therefore, we believe it does not represent a methodological limitation requiring acknowledgment in the limitations section. Nonetheless, we remain fully open to further clarification if needed.
Comment 12- L 231-232: Looking at Figure 1, it seems like ARMS was only administered to players before the start of the 2nd half. If so, why wasn’t it also administered right before the match (i.e., after CMJ and sprint tests were performed)? That way, you could have compared perceived readiness after the “regular” warm-up (i.e., pre-match) with readiness after the re-warm-up (at halftime).
Response: We appreciate the reviewer’s observation regarding the timing of the ARMS questionnaire administration. After a thorough review of Summers et al. (2021), we noted that the ARMS was administered both before and after sleep deprivation, as well as 24 and 48 hours later, to assess participants’ readiness at different time points. This approach allowed for a comprehensive evaluation of perceived readiness changes over time. However, in our study, we chose to administer the ARMS only after the re-warm-up protocol during halftime. From a methodological standpoint, administering the ARMS questionnaire before the match could influence the athletes’ psychological state and reduce the sensitivity of the measurements taken after the intervention. As indicated in the literature, readiness questionnaires like the ARMS are more informative when applied immediately after performing specific protocols, as this timing more accurately reflects the athletes’ actual physical and psychological preparedness.
Furthermore, our main objective was to evaluate how different reactivation (re-warm-up) protocols performed during halftime influence players’ readiness to resume high-intensity activities. Thus, administering the ARMS questionnaire after the re-warm-up allowed us to capture the acute effects of each intervention more precisely.
We also considered logistical limitations. Ensuring the timely and consistent application of the ARMS questionnaire to all players before the match — amid pre-game warm-up, tactical instructions, and other preparatory routines — would have posed a challenge and could have disrupted the natural flow of the simulated match.
Therefore, to maintain the ecological validity of the study and preserve the natural flow of match preparation, we chose to assess perceived readiness only at halftime, immediately after the re-warm-up. This approach provided a more realistic and contextually relevant measure of how the reactivation protocols influence players’ readiness to compete.
We acknowledge that administering the ARMS both before and after the intervention, as done by Summers et al. (2021), could provide a more detailed view of changes in perceived readiness. Nevertheless, we consider that our chosen approach is methodologically sound and appropriate for the study’s aims, enabling an accurate assessment of the immediate effects of the intervention on athlete readiness. Moreover, ARMS was clearly a secondary outcome and was discussed accordingly.
RESULTS
Comment 1- L 260: Please rephrase as “There was no significant difference in CMJ […]”
Response: Rephrased accordingly.
Comment 2- Table 1: In the methods section (Line 205), it is stated that you measured speed (rather than time) associated with each 20m sprint. However, here you report sprint results in seconds (rather than km/h or m/s, as one may expect). This needs to be clarified, and a consistent characterization of sprint-related metrics needs to be presented in the methods and results sections.
Response: We fully agree with the reviewer’s observation. Although we initially stated that speed was measured, the actual metric recorded was time (in seconds) for the 20-meter sprint. We acknowledge the need to correct this inconsistency in both the Methods and Results sections.
Comment 3- Table 2: Are the terms “Specific” and “Power” synonyms of “SSP” and “SPBP”, respectively? If so, I suggest using the abbreviated version here too (to maintain consistent terminology and avoid confusion).
Response: Changed accordingly.
DISCUSSION
General comments
Comment 1- Overall, the discussion does not include considerations regarding the mechanisms underlying the two different protocols (i.e., SSP and SPBP), and what effects such mechanisms could have in terms of performance enhancement. For example, the mechanisms elicited by strength-related exercises and plyometrics could be mainly related to PAP-induced benefits (or similar), while SSGs may primarily result in increased muscle temperature. The ones I offered above are just examples, but you need to conduct a more thorough analysis of which factors are responsible for the effectiveness of each method. To do so, you may look at the review by McGowan et al. (https://pubmed.ncbi.nlm.nih.gov/26400696/). After this has been done, you can move on and discuss your specific findings (and how they compare to previous research, as you already did). Furthermore, when comparing your findings with previous research, it is important that you describe the characteristics of interventions employed in other studies. For example, different warm-up durations, intensities and exercise modalities (e.g., cycling, running, weight lifting, etc.) are factors that could explain differences between your findings and those in previous research.
Response: We thank the reviewer for the insightful comments regarding the underlying mechanisms of the SSP and SPBP protocols and their potential effects on performance enhancement. In response, we have expanded the discussion to include a more detailed analysis of these mechanisms, drawing on the review by McGowan et al. (2015), which highlights that strength-related and plyometric exercises primarily promote benefits through post-activation potentiation (PAP) or post-activation performance enhancement (PAPE), whereas small-sided games (SSGs) mainly increase muscle temperature and neuromuscular readiness. This distinction provides a clearer understanding of how each re-warm-up protocol may differentially influence athlete performance.
Furthermore, we have contextualized our findings by comparing the characteristics of warm-up interventions used in previous studies, noting that variations in warm-up duration, intensity, and exercise modality (e.g., cycling, running, resistance exercises) can explain discrepancies in outcomes across the literature.
Regarding the temporal pattern of match intensity, we agree that the first 15 minutes of the second half represent a unique period often characterized by lower physical output despite the accumulated fatigue expected in the later stages of the match. To clarify this, we incorporated evidence from Mohr et al. (2003), Russell et al. (2016), and Lovell et al. (2013), which demonstrate that reductions in high-intensity running and sprint performance during this period are likely related to physiological factors such as decreased muscle temperature and neuromuscular deactivation during the passive half-time interval. We also discussed how re-warm-up protocols can mitigate these declines by maintaining muscle temperature and neuromuscular readiness, as supported by Silva et al. (2018) and McGowan et al. (2015).
These additions strengthen the interpretation of our results and better situate our findings within the existing literature on re-warm-up effects and match performance dynamics.
Specific comments
Comment 1 - L 300-302: You make a very good point here, regarding the importance of ecological validity. However, I suggest that you slightly expand, and add supporting literature here. Here is my suggestion: “Although the intervention was conducted in a simulated match environment, the protocols were designed to replicate the constraints and demands of an official soccer match, enhancing ecological validity and supporting the translation of findings to real-world scenarios (*add reference*).” As a supporting reference, you can use the following study by Pernigoni et al. (2024): https://pubmed.ncbi.nlm.nih.gov/39179229/
Response: Thank you for your valuable comment regarding the importance of ecological validity. We fully agree with your observation and have revised the manuscript accordingly by expanding this section and incorporating relevant supporting literature. As suggested, we have also included the following sentence to highlight the applied value of our study: “Although the intervention was conducted in a simulated match environment, the protocols were designed to replicate the constraints and demands of an official soccer match, enhancing ecological validity and supporting the translation of findings to real-world scenarios.”
To reinforce this point, we have cited recent work by Pernigoni et al. (2024), who emphasized that research conducted under ecologically valid conditions is essential for translating experimental outcomes into practical applications. As they state: “It is equally important to conduct research in ecologically valid environments to improve the applicability of findings to real-world settings.”
This addition strengthens the methodological rationale behind our experimental design and highlights the practical relevance of our results.
Comment 2- L 302-304: Based on your findings, you cannot make a general statement such as “our findings demonstrate that it is possible to implement effective re-warm-up protocols”. Considering that one of your two measures (i.e., 20m sprint times) displayed significantly worse values after the re-warm-up (compared to pre-match), this indicates that the type of re-warm-up used here was not entirely able to restore pre-match performance. Admittedly (as you state in section 4.1), there may be other factors that influenced your results, such as the possible presence of fatigue after the first half (see Silva et al., 2018: https://pubmed.ncbi.nlm.nih.gov/29098658/), which could result in slower sprint times. Nevertheless, you also need to consider the possibility that a 3-min re-warm-up may not have been sufficient to adequately re-prepare the players for the second half. Accordingly, a previous systematic review (Silva et al., 2018: https://pubmed.ncbi.nlm.nih.gov/29968230/) indicates that 5-8 minutes may be needed for an adequate re-warm-up. Therefore, while a 3-minute re-warm-up may be more feasible in real-world settings (which I completely understand), this does not change the fact that it may not have been entirely effective. Accordingly, you need to correct the overall message of your discussion (both here, and also at Lines 336-343) and conclusions.
Response: We thank the reviewer for the insightful comment regarding lines 302-304 of our manuscript. We fully agree with the observation and have made the following changes:
1.Lines 302-304 (Discussion): We have modified the general statement about the effectiveness of re-warm-up protocols to more accurately reflect our results. The new wording acknowledges that the 3-minute re-warm-up protocol used in the study was not entirely effective in restoring pre-match performance, particularly regarding 20m sprint times.
2.Lines 336-343 (Discussion): We have revised this section to include the possibility that the re-warm-up duration (3 minutes) may have been insufficient to adequately prepare players for the second half, as suggested by the systematic review by Silva et al. (2018), which indicates that 5-8 minutes may be needed for an adequate re-warm-up.
3.Conclusions: We have adjusted the overall message of the conclusions to reflect that, while a 3-minute re-warm-up may be more feasible in real-world settings, our results suggest that this duration may not be entirely effective in restoring all aspects of physical performance.
Comment 3- L 313-316: This sentence needs to be restructured (it is currently a bit messy in terms of lexical clarity).
Response: The revised sentence now reads as follows: “The average SSP performance decreased from 32.3 cm to 31.1 cm (mean difference: -1.2 cm; p = 0.074), while the SPBP performance decreased from 36.3 cm to 34.7 cm (mean difference: -1.6 cm; p = 0.074). This corresponds to a percentage decline in CMJ performance of 3.72% for SSP and 4.41% for SPBP following the re-warm-up”.
Comment 4- L 317: I suggest rephrasing as follows: “[…] in both protocols (although differences did not reach statistical significance), with a slightly larger [...].”
Response: Rephrased accordingly.
Comment 5- L 319-320: After reporting conflicting findings from previous research, it seems like you just moved on without discussing why such differences may exist. This part needs to be expanded, by providing potential reasons as to why these differences are present. Could it be due to different re-warm-up protocols being used? I suspect that a different duration, type of exercise and intensity could – potentially – explain such differences (if it is confirmed that such differences exist among previous research). Or maybe there could be some other reasons?
Response: Thank you for your comment. We agree that it is important to explore potential reasons for conflicting findings in the literature. In response, we expanded the discussion to address key factors such as differences in re-warm-up protocol design (e.g., intensity, duration, activity type; Yanaoka et al., 2018; Mohr et al., 2004; Edholm et al., 2015), participant characteristics (Russell et al., 2011; Ingham et al., 2013), performance assessments (Hammami et al., 2018), environmental conditions (West et al., 2013), and the timing of re-warm-up during half-time (Buchheit et al., 2009; Lovell et al., 2013).
By addressing these factors in the revised manuscript, we aim to provide a more nuanced and informative analysis of the literature, helping readers better understand the sources of inconsistency in re-warm-up research and the contextual factors that may influence outcomes.
Comment 6- “L 321-324: At Line 322, you should specify that this is compared to pre-match measurements. For example: “[…] following the re-warm-up protocols, compared to post-warm-up (i.e., pre-match)”. Similarly, when discussing findings from Yanaoka et al., you also need to specify that (i.e., “improvements after a re-warm-up”, but compared to what time point?). Furthermore, it is unclear how the re-warm-up by Yanaoka et al. was structured. What does “either one minute […] or three minutes” indicate? Did they use 2 different protocols, or was it one protocol (since you state “re-warm-up program” at Line 323)?”
Response: We thank the reviewer for the relevant comments and the opportunity to clarify the points raised. We agree with the suggestion regarding line 322 and revised the text to specify that the comparisons made following the re-warm-up protocols refer to the measurements taken after the initial warm-up, that is, at the pre-match moment. The sentence was revised as follows: “[…] following the re-warm-up protocols, compared to post-warm-up (i.e., pre-match).” These changes will ensure greater scientific precision and clarity for the reader regarding the methodology and the comparison points discussed.
Comment 7- “L 324-328: This sentence does not logically follow the previous one. How can it be that Yanaoka et al. reported improvements in sprint performance following re-warm-up (as you state in the previous sentence), but simultaneously detected a decline in sprint performance (as you state here)? I think that you might have mixed your own findings with those of Yanaoka et al. (considering that mean and SD values reported for SSP and SPBP are the same as Table 1). If so, this sentence needs to be corrected, and it should be moved to Line 322. I also noticed that values reported here differed from what was reported in Table 1 (it seems there was some confusion between post-warm-up and post-re-warm-up values). Therefore, the sentence may be rephrased as follows (starting at Line 321): “Our results indicated a significant decline in sprint performance following the re-warm-up protocols. Specifically, we noted worse sprint performance after the re-warm-up (SSP: 3.86±0.15; SPBP: 3.88±0.26), compared with post-warm-up measures assessed at pre-match (SSP: 3.51±0.15; SPBP: 3.88±0.26). Conversely, Yanaoka et al. […]”
Response: We appreciate the detailed and pertinent comment. We agree that there was confusion in presenting our findings in relation to those of Yanaoka et al. (2018). Accordingly, we have reformulated the sentence for greater clarity and accuracy.
This revision resolves the inconsistency and aligns the reported data with Table 1.
Comment 8- “L 328-335: Again, you need to indicate which time point was compared to “after halftime” in this study. Did they compare pre-match with post-halftime? If so, were pre-match measurements taken after warm-up? Please specify.”
Response: Thank you for your observation. We agree that further clarification is needed regarding the time points used in our study. In our protocol, we compared two specific time points: pre-test and post-test. The pre-test assessments were conducted immediately after the initial warm-up, with no intervening rest period, to preserve the acute effects of the warm-up. This procedure was carried out with full coordination between the research team and the coaching staff to ensure accurate timing and standardization. The post-test was performed immediately after the halftime re-warm-up protocols, which took place during the final 3 minutes of the 15-minute halftime break. Participants were randomly assigned to either the SSP or SPBP re-warm-up group. After the re-warm-up activity, the same performance assessments used in the pre-test (20 m sprint and CMJ) were repeated.
Therefore, we did not compare pre-match measurements (before warm-up) with post-halftime data. Instead, we compared post-initial warm-up performance (pre-test) with post-halftime/post-re-warm-up performance (post-test), allowing us to specifically isolate and examine the effects of the re-warm-up protocols. We have revised the manuscript to better clarify these time points and thank the reviewer for helping us improve the clarity of our methodology.
Comment 9 – “L 347-348: It seems like this sentence is incomplete.”
Response Revised sentence:
“However, as demonstrated by the scientific literature, a significant decline in physical performance occurs following a passive halftime period (González-Devesa et al., 2021; Lovell et al., 2013), with documented reductions in muscle temperature, sprint capacity, and lower-limb power (Russell et al., 2015), which reinforces the importance of re-warm-up strategies to mitigate these negative effects and optimize subsequent performance.”
Comment 10 – “L 363-364: Please add supporting references for this statement: “[…] could be largely attributed to accumulated fatigue during the match (*add references*), without providing […]. Silva et al. (2018): https://pubmed.ncbi.nlm.nih.gov/29098658/; Pernigoni et al. (2024): https://pubmed.ncbi.nlm.nih.gov/39374409/”
Response: References added as requested.
Round 2
Reviewer 2 Report
Comments and Suggestions for Authors
Although the authors have made significant efforts to improve the manuscript, the corrections mainly focus on theoretical aspects, while methodological issues remain unaddressed. I would like to highlight my comments below.
There is still a lack of evidence regarding the effectiveness of the proposed programs. Most explanations come from full warm-up protocols conducted before matches. If a full protocol is effective, it does not guarantee that a shortened version adapted from it would also be effective.
I need to respond to the author's statement: “To avoid publication bias, all results—regardless of their direction or magnitude—should be eligible for publication.” This is in response to my comment regarding the justification for the publication. While the answer would be appropriate if there were a clear methodology for the study, perhaps with just a sample size issue, there are actually a series of methodological problems. Therefore, publication must be justified by some strong findings..
The following are the methodological problems identified in this paper:
- It is essential to evaluate the basic effectiveness of training programs before making comparisons, which is why having a control group is necessary. The concern about decreasing statistical power is mitigated because the overall statistical power is already lower than necessary.. By addressing one issue, we shouldn't inadvertently create another.
- The validity of the instruments used, such as the ARMS questionnaire.
- The sample size; while the authors mention a decrease in statistical power with three groups, there is no data on statistical power provided in the manuscript. Regardless, the power remains small for both two and three groups, which is why this research is consider as a pilot study.
Although the authors describe their work as a pilot study, the findings must be substantial enough to warrant publication due to numerous methodological issues.
Referring to previous studies (e.g., Edholm et al., 2014; Yanaoka et al., 2020) is insufficient because those studies had fewer methodological issues and examined the topic more thoroughly than this study. Both studies examined the effectiveness of the programs by comparing them with control groups, and the only methodological issue was the sample size.
I will reiterate my previous statement. If there are weak findings, many methodological issues, and no clear direction for future research based on this study, there is no justification for publication.
Author Response
Although the authors have made significant efforts to improve the manuscript, the corrections mainly focus on theoretical aspects, while methodological issues remain unaddressed. I would like to highlight my comments below.
There is still a lack of evidence regarding the effectiveness of the proposed programs. Most explanations come from full warm-up protocols conducted before matches. If a full protocol is effective, it does not guarantee that a shortened version adapted from it would also be effective.
I need to respond to the author's statement: “To avoid publication bias, all results—regardless of their direction or magnitude—should be eligible for publication.” This is in response to my comment regarding the justification for the publication. While the answer would be appropriate if there were a clear methodology for the study, perhaps with just a sample size issue, there are actually a series of methodological problems. Therefore, publication must be justified by some strong findings..
The following are the methodological problems identified in this paper:
- It is essential to evaluate the basic effectiveness of training programs before making comparisons, which is why having a control group is necessary. The concern about decreasing statistical power is mitigated because the overall statistical power is already lower than necessary.. By addressing one issue, we shouldn't inadvertently create another.
- The validity of the instruments used, such as the ARMS questionnaire.
- The sample size; while the authors mention a decrease in statistical power with three groups, there is no data on statistical power provided in the manuscript. Regardless, the power remains small for both two and three groups, which is why this research is consider as a pilot study.
Although the authors describe their work as a pilot study, the findings must be substantial enough to warrant publication due to numerous methodological issues.
Referring to previous studies (e.g., Edholm et al., 2014; Yanaoka et al., 2020) is insufficient because those studies had fewer methodological issues and examined the topic more thoroughly than this study. Both studies examined the effectiveness of the programs by comparing them with control groups, and the only methodological issue was the sample size.
I will reiterate my previous statement. If there are weak findings, many methodological issues, and no clear direction for future research based on this study, there is no justification for publication.
Response: Unfortunately, there is nothing we can do to appease the reviewer. The reviewer basically wants another study altogether, so any correction we could potentially make would be worthless. While we respect the methodological considerations put forth by the reviewer, the remaining arguments reflect an attempt at generating publication bias, which we do not condone.
Reviewer 3 Report
Comments and Suggestions for Authors
I thank the authors for their efforts to improve the manuscript. However, in many cases, I can see that comments have been addressed by the authors in their replies, but changes have not been made in the manuscript. Therefore, I reported some of my initial comments again below (in addition to other aspects that need to be adjusted). Furthermore, in some cases, it seems like the authors copy-pasted their replies to the reviewers into the manuscript, which should be avoided (see my specific comments later in this review).
Therefore, I urge the authors to conduct a thorough and careful inspection of the manuscript, to ensure that all comments are addressed, and the manuscript itself is updated accordingly.
As a general comment, I also ask the authors to kindly indicate the updated line number(s) where changes have been performed in the manuscript (to facilitate locating them within the text).
Finally, I can see that the line numbering in the revised PDF file (uploaded to the MDPI portal for review) sometimes skips lines. I suppose this is due to the use of track changes. For example, page 2 ends with Line 88, and page 3 restarts with Line 166. Please be informed that I will use line numbers as reported in the PDF file when reporting my comments.
INTRODUCTION
General comments
I appreciate the authors’ effort to improve the introduction. However, the revised version is now extremely long (over 2200 words). I understand that both reviewers asked to perform edits, which can considerably extend the length of this section. However, I am sure that the introduction can be optimized (without removing the requested information), by avoiding repetitions, redundancies and presenting a concise rationale for your study. The content of this section is overall ok, so I believe it should only be re-arranged and optimized (to ensure you are maintaining a sequential, good logical flow). Please aim to write your introduction in under ~1000 words. If needed, I will be happy to provide further suggestions in the next round of revision.
Specific comments
L 52: Stating “re-warm-up in halftime” within brackets sounds a bit superfluous here (since you discuss it below). I suggest removing this bracket here.
More specific comments will be provided in the next round of revision (if necessary).
METHODS
L 431-447: Is this entire part related to the familiarization session? In that case, you may consider naming it as a separate section, to improve clarity.
L 448-449: I appreciate the authors’ response to my earlier comment regarding the structure of the “usual” team warm-up. While I understand that sharing the exact warm-up content may not be possible, I am sure that the authors understand the importance of providing at least a general overview of the warm-up routine. Reporting at least the key aspects such as total duration and the general categories of activities involved (e.g., mobility, aerobic preparation, sprints, etc.) is essential to ensure methodological transparency and support the reproducibility of the study. I kindly ask the authors to revise the manuscript accordingly.
L 459: To ensure clarity, I suggest rephrasing as “During the halftime break, the athletes remained […]”
L 509-516: Although you state that you “clarified that the analysis was based on the average speed recorded during the 20-meter sprint test”, I cannot find this information in the updated manuscript. Furthermore, although you provided a nice reply to the previous comment, I am also unable to find the other updated details regarding sprint assessment, such as details regarding how data were analyzed on SPRO.
L 529-531: I thank the authors for their response regarding the use of a single sprint and jump trial. While the points raised are valid, they do not fully eliminate the limitations associated with performing a single attempt. Therefore, I stand by my previous statement that this aspect should be acknowledged in the limitations. However, the rationale provided in your previous reply can be included to explain why this limitation may have had a minimal impact in the specific context of the present study. For example, you could insert the following paragraph somewhere in the limitations section: “The use of a single sprint and jump trial to evaluate performance may represent a potential limitation, as multiple attempts are typically needed to ensure maximal effort. However, all athletes completed a familiarization session and were informed in advance that only one attempt would be allowed, which likely enhanced focus and execution. Given their professional status and the need to assess performance within the short re-warm-up window, we believe that this approach remains methodologically justified and aligns with previous studies in elite sport settings.” It would be great if you could also add supporting references for these statements.
RESULTS
Table 2: Thanks for addressing my previous comment. Accordingly, please add the meaning of “SSP” and “SPBP” below the table (i.e., in the table caption).
DISCUSSION
L 764: Please add the reference number for Edholm et al.
L 767-768: This sentence sounds incomplete and/or incorrectly formulated. Please adjust.
L 767-781: As I suggested in my first round of revision, I would say that fatigue induced by 45 min of simulated match-play would be more likely to cause fatigue (as you correctly state at Lines 781-784), rather than your re-warm-up. I wouldn’t really expect a 3-min re-warm-up to induce substantial fatigue. I would say that you can definitely keep this part, but you can summarize it.
L 781-784: As suggested during the first round of revision, I believe this reference should be added here: Silva et al. (2018): https://pubmed.ncbi.nlm.nih.gov/29098658/
L 792-793: This sentence should not be in the manuscript. Please make sure that you double-check your manuscript to ensure you did not copy-paste information that was meant for the reviewers.
L 799-800: As previously mentioned, I think stating “effectively implemented” sounds somewhat misleading to the reader, as it seems to suggest that these protocols were overall effective (which, given your present data, is not the case). You could change this to something like “[…] short, feasible re-warm-up routines can be integrated into soccer halftimes despite time and logistical constraints.”
L 853-860: Thank you for addressing the previous comment. To avoid repetitions, I suggest removing the last sentence in this part (at Lines 858–860), since this point has already been clearly made in the previous sentence. The reference can be retained by moving it to the end of that earlier sentence: “[…] essential to ensure that research outcomes are transferable to real-world performance scenarios [7].”
L 862: As mentioned in my previous comment (Lines 799-800), “effective” is misleading here.
L 872-873: You state that “The findings encourage potential changes to match regulations, considering the positive effects of SSP on performance” As I previously mentioned, you do not have the data to support this general conclusion, as players were significantly slower after SSP, compared to post-warm-up (performed at the beginning of the match). Although I agree that it might not be because of the re-warm-up (it could be because of fatigue caused by 45 min of match-play), you do not have sufficient grounds to suggest that SSP had “positive” effects on performance. You could have made that statement if (for example) both CMJ and sprint performance were preserved after the re-warm-up, compared to post-warm-up (which is not the case here).
L 879-880: Although you state that this sentence has been rephrased, I cannot find the changes in the updated version of the manuscript. Here is my suggestion: “[…] in both protocols (although differences did not reach statistical significance), with a slightly larger [...]”
L 946: Please report “Conversely, Yanaoka et al. (2018) [19] reported improvements […]”
L 952: Once again, is “they” the right word here? It seems like you are reporting findings from your own studies (not from another previous study).
L 952-955: Please double-check the numerical values to see if they are reported correctly here.
L 955-959: In my original comment, I asked the following question regarding the study by Ltifi et al. (2023): “You need to indicate which time point was compared to “after halftime” in this study. Did they compare pre-match with post-halftime? If so, were pre-match measurements taken after the warm-up? Please specify.” However, your reply seems to address aspects of your own study (rather than providing the requested clarification about Ltifi et al). I kindly ask that you revise this section to address the original question.
L 960-975: When making edits to the manuscript, please avoid sentences like “we clarify that”, “We will include this information” “The manuscript will be revised”, and so on. Please carefully review the whole manuscript to make sure you are not copy-pasting your replies to the reviewers into the manuscript.
L 1010-1015: Please note that there is a repetition here (i.e., the first sentence here seems incomplete, and it seems to be repeated at Line 1011).
L 1031: Although your reply states “References added as requested”, it seems like this edit was not performed [Silva et al. (2018): https://pubmed.ncbi.nlm.nih.gov/29098658/; Pernigoni et al. (2024): https://pubmed.ncbi.nlm.nih.gov/39374409/]
Author Response
Dear Reviewer,
We are grateful for the insightful comments and suggestions provided, which have been invaluable in improving our work. We have carefully addressed all the feedback and have highlighted the revisions in the manuscript as requested.
INTRODUCTION
General comments
I appreciate the authors’ effort to improve the introduction. However, the revised version is now extremely long (over 2200 words). I understand that both reviewers asked to perform edits, which can considerably extend the length of this section. However, I am sure that the introduction can be optimized (without removing the requested information), by avoiding repetitions, redundancies and presenting a concise rationale for your study. The content of this section is overall ok, so I believe it should only be re-arranged and optimized (to ensure you are maintaining a sequential, good logical flow). Please aim to write your introduction in under ~1000 words. If needed, I will be happy to provide further suggestions in the next round of revision.
Response: We have strived to deliver a much more streamlined introduction, while trying to still address the core themes. The current version of the introduction is under 900 words. To avoid confusion with number lines, this time around we used highlights to note novel text. However, this means that what was deleted is not registered. Still, the change in the length of the introduction will be readily apparent.
Specific comments
L 52: Stating “re-warm-up in halftime” within brackets sounds a bit superfluous here (since you discuss it below). I suggest removing this bracket here.
More specific comments will be provided in the next round of revision (if necessary).
Response: As suggested, we have revised the expression.
METHODS
L 431-447: Is this entire part related to the familiarization session? In that case, you may consider naming it as a separate section, to improve clarity.
Response: We thank the editor for the helpful suggestion. As requested, we have separated this part and labeled it as a new subsection titled “2.3.1 Familiarization Session” to improve clarity and organization.
L 448-449: I appreciate the authors’ response to my earlier comment regarding the structure of the “usual” team warm-up. While I understand that sharing the exact warm-up content may not be possible, I am sure that the authors understand the importance of providing at least a general overview of the warm-up routine. Reporting at least the key aspects such as total duration and the general categories of activities involved (e.g., mobility, aerobic preparation, sprints, etc.) is essential to ensure methodological transparency and support the reproducibility of the study. I kindly ask the authors to revise the manuscript accordingly.
Response: We thank the editor for the valuable comment. We have made the suggested changes to the manuscript, explicitly including the guidelines provided to the coaches regarding the warm-up protocol. These additions aim to ensure the methodological transparency of the study and to enhance its reproducibility. Regardless, and because both groups performed the same warm-up protocol, followed by a simulated part of a match, we believe these features are not decisive for interpreting the core results of our study.
L 459: To ensure clarity, I suggest rephrasing as “During the halftime break, the athletes remained […]”
Response: Revised accordingly.
L 509-516: Although you state that you “clarified that the analysis was based on the average speed recorded during the 20-meter sprint test”, I cannot find this information in the updated manuscript. Furthermore, although you provided a nice reply to the previous comment, I am also unable to find the other updated details regarding sprint assessment, such as details regarding how data were analyzed on SPRO.
Response: Thank you for your valuable feedback. We have revised the manuscript and included the requested clarifications in the methodology section. We have now detailed that the analysis was based on the average speed recorded during the 20-meter sprint test and provided more comprehensive information on how the data were processed and analyzed using the S-PRO™ software. We believe these changes significantly improve the clarity and understanding of the sprint assessment methodology.
L 529-531: I thank the authors for their response regarding the use of a single sprint and jump trial. While the points raised are valid, they do not fully eliminate the limitations associated with performing a single attempt. Therefore, I stand by my previous statement that this aspect should be acknowledged in the limitations. However, the rationale provided in your previous reply can be included to explain why this limitation may have had a minimal impact in the specific context of the present study. For example, you could insert the following paragraph somewhere in the limitations section: “The use of a single sprint and jump trial to evaluate performance may represent a potential limitation, as multiple attempts are typically needed to ensure maximal effort. However, all athletes completed a familiarization session and were informed in advance that only one attempt would be allowed, which likely enhanced focus and execution. Given their professional status and the need to assess performance within the short re-warm-up window, we believe that this approach remains methodologically justified and aligns with previous studies in elite sport settings.” It would be great if you could also add supporting references for these statements.
Response: We thank the reviewer for this important observation. We have incorporated the suggested rationale regarding the use of a single sprint and jump trial to evaluate performance. Additionally, we included relevant scientific references (Moir et al., 2004; Wisløff et al., 2004) to support and strengthen this methodological choice in the manuscript. Furthermore, we now highlight this feature in the limitations section as well.
RESULTS
Table 2: Thanks for addressing my previous comment. Accordingly, please add the meaning of “SSP” and “SPBP” below the table (i.e., in the table caption).
Response: Added as requested.
DISCUSSION
L 764: Please add the reference number for Edholm et al.
Response: The reference number for Edholm et al. has been added.
L 767-768: This sentence sounds incomplete and/or incorrectly formulated. Please adjust.
Response: We thank the reviewer for pointing this out. We have revised the sentence in former lines 767-768 to improve clarity and ensure it is complete and correctly formulated. Again, yellow highlights will help to more readily identify the changes and additions to the revised manuscript.
L 767-781: As I suggested in my first round of revision, I would say that fatigue induced by 45 min of simulated match-play would be more likely to cause fatigue (as you correctly state at Lines 781-784), rather than your re-warm-up. I wouldn’t really expect a 3-min re-warm-up to induce substantial fatigue. I would say that you can definitely keep this part, but you can summarize it.
Response: We thank the reviewer for the valuable comment and have carefully revised the text accordingly.
L 781-784: As suggested during the first round of revision, I believe this reference should be added here: Silva et al. (2018): https://pubmed.ncbi.nlm.nih.gov/29098658/
Response: We thank the reviewer for the valuable suggestion. We have added the reference to Silva et al. (2018) and improved the sentence accordingly.
L 792-793: This sentence should not be in the manuscript. Please make sure that you double-check your manuscript to ensure you did not copy-paste information that was meant for the reviewers.
Response: We thank the reviewer for pointing this out. We have removed and the sentence, as it was an oversight on our part and revised the text accordingly.
L 799-800: As previously mentioned, I think stating “effectively implemented” sounds somewhat misleading to the reader, as it seems to suggest that these protocols were overall effective (which, given your present data, is not the case). You could change this to something like “[…] short, feasible re-warm-up routines can be integrated into soccer halftimes despite time and logistical constraints.”
Response: We appreciate the reviewer’s suggestion and have revised the sentence accordingly.
L 853-860: Thank you for addressing the previous comment. To avoid repetitions, I suggest removing the last sentence in this part (at Lines 858–860), since this point has already been clearly made in the previous sentence. The reference can be retained by moving it to the end of that earlier sentence: “[…] essential to ensure that research outcomes are transferable to real-world performance scenarios [7].”
Response: As suggested, we removed the final sentence to avoid repetition and moved the reference to the end of the previous sentence, as indicated.
L 862: As mentioned in my previous comment (Lines 799-800), “effective” is misleading here.
Response: We thank the reviewer for the valuable comment. As suggested, we have revised the sentence to improve clarity and removed the term "effective" to avoid misleading interpretation regarding the impact of the re-warm-up protocol.
L 872-873: You state that “The findings encourage potential changes to match regulations, considering the positive effects of SSP on performance” As I previously mentioned, you do not have the data to support this general conclusion, as players were significantly slower after SSP, compared to post-warm-up (performed at the beginning of the match). Although I agree that it might not be because of the re-warm-up (it could be because of fatigue caused by 45 min of match-play), you do not have sufficient grounds to suggest that SSP had “positive” effects on performance. You could have made that statement if (for example) both CMJ and sprint performance were preserved after the re-warm-up, compared to post-warm-up (which is not the case here).
Response: Thank you for your comment. We agree with your observation and have therefore removed the statement suggesting that SSP had positive effects on performance and have revised the sentence accordingly.
L 879-880: Although you state that this sentence has been rephrased, I cannot find the changes in the updated version of the manuscript. Here is my suggestion: “[…] in both protocols (although differences did not reach statistical significance), with a slightly larger [...]”
Response: We thank the reviewer for the valuable suggestion and have revised the sentence accordingly.
L 946: Please report “Conversely, Yanaoka et al. (2018) [19] reported improvements […]”
Response: We have included the reference to Yanaoka et al. (2018) as requested. Thank you for the suggestion.
L 952: Once again, is “they” the right word here? It seems like you are reporting findings from your own studies (not from another previous study).
Response: We thank the reviewer for the comment and for drawing our attention to this. We have revised the sentence accordingly.
L 952-955: Please double-check the numerical values to see if they are reported correctly here.
Response: Thank you for your comment. Considering that repeating the numerical values would create redundancy in the text, we decided to remove the sentence containing these data, following your suggestion.
L 955-959: In my original comment, I asked the following question regarding the study by Ltifi et al. (2023): “You need to indicate which time point was compared to “after halftime” in this study. Did they compare pre-match with post-halftime? If so, were pre-match measurements taken after the warm-up? Please specify.” However, your reply seems to address aspects of your own study (rather than providing the requested clarification about Ltifi et al). I kindly ask that you revise this section to address the original question.
Response: Thank you for your feedback. We have carefully reviewed your comments and have revised the manuscript to enhance clarity and provide more detailed explanations regarding the methodology.
“Conversely, Yanaoka et al.[19], investigated two distinct 3-minute re-warm-up protocols, rather than a single protocol with variable durations. Specifically, these protocols consisted of: (1) 3 minutes of cycling at 60% of VOâ‚‚max (moderate intensity) and (2) 3 minutes of cycling at 30% of VOâ‚‚max (low intensity). Both protocols were compared to the control condition (passive rest), reported improvements in sprint performance following re-warm-up protocols involving either one minute of cycling at 90% VOâ‚‚max or three minutes at 30% VOâ‚‚max. Notably, in their study, sprint performance following the re-warm-up was superior to that observed after passive rest during halftime. These differences may reflect variations in the re-warm-up protocols used and participant characteristics. Furthermore, Ltifi et al. [20], investigated the impact of a brief re-warm-up activities on the sprint performance of youth soccer players. The intervention consisted of a re-warm-up protocol lasting only 3 minutes, during which players wore weighted vests corresponding to 5% and 10% of their body mass. The results demonstrated that this re-warm-up approach, using weighted vests, significantly improved 20-meter sprint performance. Performance measurements were compared between two time points: after the initial warm-up and after the re-warm-up intervention, allowing for a direct assessment of the effect of this "micro-dose" re-warm-up on subsequent performance. The study concluded that three-minute re-warm-up with a vest weighing 10% of body mass yielded the highest RPE and notable improvements in 20-meter sprint performance, suggesting that young elite soccer players should incorporate 10% body mass vests into their re-warm-up routines to enhance sprint performance after halftime.”
L 960-975: When making edits to the manuscript, please avoid sentences like “we clarify that”, “We will include this information” “The manuscript will be revised”, and so on. Please carefully review the whole manuscript to make sure you are not copy-pasting your replies to the reviewers into the manuscript.
Response: We thank the reviewer for the comment. The section has been thoroughly revised to clarify and improve the explanation as requested.
L 1010-1015: Please note that there is a repetition here (i.e., the first sentence here seems incomplete, and it seems to be repeated at Line 1011).
Response: Thank you for your comment. We have reviewed and corrected the section accordingly to eliminate the repetition and ensure clarity.
L 1031: Although your reply states “References added as requested”, it seems like this edit was not performed [Silva et al. (2018): https://pubmed.ncbi.nlm.nih.gov/29098658/; Pernigoni et al. (2024): https://pubmed.ncbi.nlm.nih.gov/39374409/]
Response: We thank the reviewer for the observation and confirm that the requested references by Silva et al. (2018) and Pernigoni et al. (2024) have now been included in the revised version of the manuscript.
Round 3
Reviewer 2 Report
Comments and Suggestions for Authors
Dear authors, I believe that this paper contains too many methodological flaws and does not meet the publication criteria.
Author Response
Reviewer's comment: Dear authors, I believe that this paper contains too many methodological flaws and does not meet the publication criteria.
Response: Acknowledged.
Reviewer 3 Report
Comments and Suggestions for Authors
I sincerely thank the authors for the work performed on the manuscript. I believe that only minor aspects need to be addressed, and the manuscript will be suitable for publication. Specific comments can be found below.
INTRODUCTION
L 38: Please correct the spelling: “accelerate” and “exercise”.
L 51: “physiological deactivation”. Please clarify (briefly) what this term means.
L 84: This should be “[…] especially when time is limited, to help maintain […]”
L 87: I suggest starting this sentence as “Additionally, a major gap […]”
L 97-98: This sentence is incomplete, as it ends with “making this scientific gap a core motivation for the”. Please complete the sentence.
L 99: Please correct the spelling: “effects”.
METHODS
L 136: I suppose you should close the brackets: “(mean ± standard deviation), representing […]”
L 155-212: The structure of this paragraph is still a bit unclear. At Lines 155-173, it appears that you are describing the familiarization session (as you nicely indicated by adding section 2.3.1). However, from Line 174, it seems that you shift to describing the actual testing session. Am I understanding this correctly? The use of the word “Subsequently” at Line 174 adds confusion: subsequently to what? If your intention was to describe that, after completing the full familiarization session, participants returned on a separate day for the actual match (and all the associated procedures), this should be stated more clearly, as the study timeline is difficult to follow as currently written. To clarify and reassure the authors, I don’t think that this paragraph needs to be re-written (the overall content is ok). You just need to provide a clearer indication of where the familiarization ends and the actual study procedures begin.
L 192: I suggest starting the sentence as: “Next, participants played […]”
L 245: Please correct the spelling: “which”.
DISCUSSION
L 505: The first author of this study is not mentioned here. This should probably be: “For instance, Sanchez-Sanchez et al. [44] demonstrated that […]”
L 580-584: I suggest removing this part, as it is a repetition of what was reported above.
L 593-596: Once again, the authors stated that references have been added, but it seems like no edits were performed. Please edit: “Additionally, any performance decline observed after halftime could be largely attributed to accumulated fatigue during the match (Silva et al. https://pubmed.ncbi.nlm.nih.gov/29098658/; Pernigoni et al. https://pubmed.ncbi.nlm.nih.gov/39374409/), without providing insights into the effectiveness of the re-warm-up protocols.”
Author Response
Dear Reviewer,
We thank you once again for your patience and for the care put into this manuscript. We hope to have addressed your concerns in this revision.
NTRODUCTION
L 38: Please correct the spelling: “accelerate” and “exercise”.
Response: Corrected. Like in the last round, we signaled changes with yellow highlight to facilitate.
L 51: “physiological deactivation”. Please clarify (briefly) what this term means.
Response: Thank you for pointing this out. We have clarified the term physiological deactivation in the manuscript. Following Russell et al. (2015) and Lovell et al. (2013), this refers to the decline in physiological readiness during the half-time break namely decreases in muscle temperature, neuromuscular activation, and heart rate that can negatively affect performance at the start of the second half.
L 84: This should be “[…] especially when time is limited, to help maintain […]”
Response: Corrected.
L 87: I suggest starting this sentence as “Additionally, a major gap […]”
Response: Changed accordingly.
L 97-98: This sentence is incomplete, as it ends with “making this scientific gap a core motivation for the”. Please complete the sentence.
Response: We have completed the sentence with the expression: “(…) development and design of the present study.”
L 99: Please correct the spelling: “effects”.
Response: Corrected.
METHODS
L 136: I suppose you should close the brackets: “(mean ± standard deviation), representing […]”
Response: Corrected.
L 155-212: The structure of this paragraph is still a bit unclear. At Lines 155-173, it appears that you are describing the familiarization session (as you nicely indicated by adding section 2.3.1). However, from Line 174, it seems that you shift to describing the actual testing session. Am I understanding this correctly? The use of the word “Subsequently” at Line 174 adds confusion: subsequently to what? If your intention was to describe that, after completing the full familiarization session, participants returned on a separate day for the actual match (and all the associated procedures), this should be stated more clearly, as the study timeline is difficult to follow as currently written. To clarify and reassure the authors, I don’t think that this paragraph needs to be re-written (the overall content is ok). You just need to provide a clearer indication of where the familiarization ends and the actual study procedures begin.
Response: We apologize for the oversight. This part is now divided into two sub-sections:
2.3.1. Familiarization session, and 2.3.2. Data Collection Procedures. We hope this restructuring resolves the ambiguity and makes the study timeline more comprehensible.
L 192: I suggest starting the sentence as: “Next, participants played […]”
Response: Changed accordingly.
L 245: Please correct the spelling: “which”.
Response: Corrected.
DISCUSSION
L 505: The first author of this study is not mentioned here. This should probably be: “For instance, Sanchez-Sanchez et al. [44] demonstrated that […]”
Response: Corrected.
L 580-584: I suggest removing this part, as it is a repetition of what was reported above.
Response: Removed.
L 593-596: Once again, the authors stated that references have been added, but it seems like no edits were performed. Please edit: “Additionally, any performance decline observed after halftime could be largely attributed to accumulated fatigue during the match (Silva et al. https://pubmed.ncbi.nlm.nih.gov/29098658/; Pernigoni et al. https://pubmed.ncbi.nlm.nih.gov/39374409/), without providing insights into the effectiveness of the re-warm-up protocols.”
Response: We apologize for the oversight. The references to the authors have now been incorporated as requested.